biomathematics/mathematical modelling/
computational biology

SARS-CoV-2, innate immune response,
viral dynamics, within-host

**Author for correspondence:**
R. N. Leander
e-mail: rachel.leander@mtsu.edu

# A model of the innate immune response to SARS-CoV-2 in the alveolar epithelium

R. N. Leander[1], Y. Wu[1], W. Ding[1], D. E. Nelson[2] and Z. Sinkala[1]

[1]Department of Mathematical Sciences, and [2]Department of Biology, Middle Tennessee State University, Murfreesboro 37132-0002, USA

RNL, 0000-0002-1392-9754; DEN, 0000-0002-5034-0885

We present a differential equation model of the innate immune response to SARS-CoV-2 within the alveolar epithelium. Critical determinants of the viral dynamics and host response, including type I and type II alveolar epithelial cells, interferons, chemokines, toxins and innate immune cells, are included. We estimate model parameters, compute the within-host basic reproductive number, and study the impacts of therapies, prophylactics, and host/pathogen variability on the course of the infection. Model simulations indicate that the innate immune response suppresses the infection and enables the alveolar epithelium to partially recover. While very robust antiviral therapy controls the infection and enables the epithelium to heal, moderate therapy is of limited benefit. Meanwhile interferon therapy is predicted to reduce viral load but exacerbate tissue damage. The deleterious effects of interferon therapy are especially apparent late in the infection. Individual variation in ACE2 expression, epithelial cell interferon production, and SARS-CoV-2 spike protein binding affinity are predicted to significantly impact prognosis.

## 1. Introduction

Since the emergence of SARS-CoV-2 in December 2019, numerous epidemiological between-host models have been developed to forecast the spread of the virus across the USA and globally [1–9]. These have predicted the effects of non-pharmaceutical interventions and influenced government policy. By comparison, the use of mathematical modelling to investigate the within-host dynamics of the virus have been less common [10–12]. These models typically simulate the infection of a homogeneous

population of cells and are parametrized by patient pathology data, such as chest radiograph score [10], or viral load [11,12]. Within-host models are useful tools for exploring how the course of infection is influenced by model parameters, especially those related to the host's immune system and pharmaceutical interventions.

One very early within-host model was limited by the availability of patient data for model fitting, requiring the use of chest radiographic score as a proxy for the infection of epithelial cells [10]. Despite this limitation, by fitting a simple three-compartment model (including target cells, infectious cells, and virus) to data, Li *et al.* calculated a within-host basic reproductive number ($R_0$) of 3.79 for SARS-CoV-2 and compared this with the $R_0$ value estimated previously for the closely related and more virulent MERS-CoV ($R_0$ of 8.16) [10]. In addition, [10] explored the sensitivity of $R_0$ to model parameters, including those related to host immunity, and numerically simulated the impact of antiviral treatment. It is important to note that the model presented in [10] does not include a dynamic immune response, instead the host's immunity is completely determined by static model parameters. By contrast, [11] compared the ability of three simple models to describe viral load data. Two of these were target-cell-limited models, which, similar to that in [10], lacked a dynamic immune response. The third was a two-compartment model, which included a dynamic immune response but neglected host cell dynamics. The two-compartment model was determined to best fit viral load data. The within-host basic reproductive number was also computed for several patients by fitting the target-cell-limited model to data. The basic reproductive number varied widely between patients, but was predicted to be very large, with a mean value greater than 10.

More recently, Wang *et al.* [12] used viral load data from human patients [13–15] and primate studies [16] to explore how the course of infection and viral load dynamics are shaped by cell-mediated and humoral immune responses. Here, the cell-mediated response is proportional to the concentration of lymphocytes present, and the humoral response leads to an exponential increase in the rate of virus decay later in the infection. Simulations of the model indicated that while the cell-mediated response is able to bring viral load down to a low plateau, the humoral response is necessary for clearance of the virus. This work also considered how decreasing the rate of viral replication and increasing the rate of cell-mediated killing might impact the course of the infection, as these changes coarsely represent the effect of anti-viral and interferon treatment, respectively.

At time of writing, the worldwide distribution of vaccine has recently begun. This campaign will take many months, if not years, to complete, and it remains unclear whether it will successfully bring the current COVID-19 pandemic to a close. Hence, there remains a need to better understand the within-host dynamics of SARS-CoV-2 infection and how these dynamics are shaped by the immune system and therapeutic interventions, alike. Here, we present a model of the innate immune response to SARS-CoV-2 infection within the alveolar epithelium. Although, SARS-CoV-2 infects a wide range of tissues [17], infection of the lung plays a central role in disease progression, with pneumonia and acute respiratory distress syndrome (ARDS) being major complications [18]. Similarly, while the adaptive immune response is doubtless critical for determining the ultimate outcome of infection, we specifically focus our model on the initial innate immune response in order to (i) better understand factors that influence the propensity of the virus to take root in the alveolar epithelium, and (ii) determine how the scene is set when the adaptive immune response is mustered. In this way, our model differs significantly from those that precede it, accounting for both the unique biology of the alveolar epithelial cells and the distinctive environment/architecture they create. In particular, our model includes the demographics of these cells and their varied susceptibility to infection. Also, we model the alveolar epithelium as a surface surrounded by a thin layer of fluid. Our concrete representation of the epithelial structure enables us to better estimate model parameters (e.g. phagocytic parameters) from available empirical data, where previous models relied heavily on model fitting for parameter estimation. Finally, our model provides an explicit description of the interferon response and the recruitment and activation of innate immune cells, including how these processes contribute to viral control and tissue damage. This level of detail imparts predictive power. For example, the model could be useful for predicting how new strains of the virus might behave in the lung, studying pharmaceutical interventions and exploring the impact of patient variability on infection dynamics.

In §2, we present the model with and without disease, along with a characterization of the model steady states and basic reproductive number. Additional support for the model and details on model parametrization are provided in appendix A. In §3, we numerically simulate the model in order to fit the more uncertain parameters within the ranges delineated in appendix A and also perform numerical simulations to investigate how pharmaceutical interventions, prophylactic measures and individual variability impact the dynamics of the infection. Finally, in §4, we summarize our results.

# 2. The model

## 2.1. Model without virus

### 2.1.1. Model overview

The alveolar epithelium forms a barrier between the lung and the outside air. Our model of the alveolar epithelium includes alveolar type I cells, alveolar type II cells and alveolar macrophages. Thin alveolar epithelial type I cells mediate oxygen exchange, while alveolar epithelial type II cells maintain the integrity of the alveolar region, in part through the secretion of lung surfactant [19]. In addition, alveolar type II cells proliferate and differentiate into type I cells [19]. A fraction of alveolar type II cells, termed alveolar epithelial progenitor cells (AEPs), are thought to be responsible for maintaining homeostasis of the healthy lung [20,21]. In response to lung injury, AEPs and other alveolar type II cells proliferate [21,22]. Indeed, whereas only about a small fraction of alveolar type II cells show positive markers of proliferation in the healthy lung, after severe lung injury 85% of these cells show active proliferation [21].

Alveolar type II cells are further characterized by their expression of angiotensin-converting enzyme 2 (ACE2), the receptor that mediates entry of SARS-CoV-2. Indeed, alveolar type II cells are the primary ACE2-expressing cell in the alveolus [23]. Nonetheless, only a small fraction, about 1–7%, of alveolar type II cells are thought to express ACE2 [23,24]. Interestingly, some evidence suggests that ACE2 may be specifically enriched in AEP type II cells [23,24]. Similar to AEPs, neural progenitor cells were found to express ACE2 [25]. The observation that ACE2 is expressed by multiple progenitor cell types, together with the fact that ACE2 expression is specifically required for exercise-induced neural proliferation [26], suggests that ACE2 may promote proliferation of alveolar epithelial cells. However, this idea is at odds with a general paradigm in which ACE2 opposes renin angiotensin system-mediated proliferation [27]. Indeed, ACE2 was found to be specifically depleted in actively proliferating, fibrotic regions of the human lung [28]. On balance, research on ACE2 expression and cellular proliferation leads us to conclude that ACE2 participates in balancing a tissue's proliferative response, possibly opposing or promoting proliferation in a context-dependent manner. In particular, we do not find sufficient evidence to support a model in which ACE2 expression either indicates or contradicts proliferation of alveolar type II cells. Hence we assume that ACE2-positive (ACE2$^+$) and ACE2-negative (ACE2$^-$) alveolar type II cells are equally likely to proliferate.

ACE2 is a dynamically regulated component of the renin angiotensin system, and its activity is partially determined by the balance of signalling through opposing arms of this system [29]. SARS-CoV-2 infection specifically downregulates ACE2 expression on the host cell's surface by inducing ACE2 endocytosis and stimulating enzymes that cleave ACE2 [29]. Meanwhile, ACE2 transcription may be promoted by interferon signalling and c-Jun N-terminal kinase (JNK) activation [28,30]. As ACE2 expression is believed to be a primary determinant of susceptibility, the dynamic regulation of ACE2 supports a model in which cells actively transition between the susceptible and immune classes. Although changes in the extracellular environment due to infection of surrounding cells may alter movement between these classes through a myriad of mechanisms, for simplicity, we focus on interferon stimulation and infection as the primary mechanisms through which susceptible cells are actively depleted, and let the basal rates of transition between ACE2$^+$ and ACE2$^-$ classes remain unchanged by infection. Note then, the ratio of the latter rates is determined by the fraction of cells which are susceptible/ACE2$^+$.

Let $A_1(t)$, $A_2^-(t)$ and $A_2^+(t)$ be the numbers of alveolar type I cells, immune (ACE2$^-$) alveolar type II cells and susceptible (ACE2$^+$) alveolar type II cells, measured in units of millions of cells. Our equations to describe the dynamics of the alveolar cells are as follows:

$$\left.\begin{aligned}
\frac{dA_1}{dt} &= aA_2 - \sigma_A A_1, \\
\frac{dA_2^-}{dt} &= r_2\left(1 - \frac{A_1 + A_2}{K_A}\right)A_2^- + a_2^+ A_2^+ - (a + a_2^- + \sigma_A)A_2^-
\end{aligned}\right\} \quad (2.1)$$

and

$$\frac{dA_2^+}{dt} = r_2\left(1 - \frac{A_1 + A_2}{K_A}\right)A_2^+ + a_2^- A_2^- - (a + a_2^+ + \sigma_A)A_2^+,$$

where we let the reproduction of type II cells follow a logistic type equation with growth rate $r_2$ and carrying capacity $K_A$, and where $A_2 = A_2^+ + A_2^-$ is the total number of type II cells. The mortality rate

of both cells types is $\sigma_A$. The rate of differentiation, $a$, is

$$a = \delta\left(1 - \frac{A_1}{K_{A1}}\right),$$

where $K_{A1}$ controls the fraction of type II cells that are differentiating as a function of the number of type I cells present, and $\delta$ is the maximal average rate of differentiation. Finally, type II cells transition from immune to susceptible and back with rates $a_2^- = \gamma p_+(\delta + \sigma_A)$ and $a_2^+ = \gamma(1 - p_+)(\delta + \sigma_A)$, so that when the ACE2-expression structure of the $A_2$ cell population equilibrates, a fraction, $p_+$, of these cells will be positive for the ACE2 receptor. The factor $\gamma$ is chosen to be at least one so that the timescale on which the structure of the population equilibrates is determined relative to the type II cell's lifespan. In particular, the ACE2 expression structure is expected to equilibrate fairly quickly, since the ACE2 receptor is important for maintaining appropriate pulmonary pressure [31].

Finally, we let $M(t)$ be the population of inactivated immune cells in the alveolar region. We suppose these cells are recruited at a constant rate $r_M$ and die at a constant *per capita* rate $\sigma_M$. Then the dynamics of the inactivated immune cells in the absence of infection are described as follows:

$$\frac{dM}{dt} = r_M - \sigma_M M.$$

### 2.1.2. Characterization of the model steady states

We find that model (2.1) has a trivial steady state and a unique positive steady state, $(\bar{A}_1, \bar{A}_2^-, \bar{A}_2^+)$, such that

— $0 < \bar{A}_1 < K_{A1}$ and
— $0 < \bar{A}_1 + \bar{A}_2^+ + \bar{A}_2^- < K_A$ provided $r_2 > \delta + \sigma_A$ and $\delta/K_{A1} < r_2/K_A$.

Note that in biological terms, the final two conditions assert respectively that cells proliferate on a faster timescale than they differentiate and die, and that the rate of differentiation is less sensitive to changes in the size of the type I cell population than the rate of proliferation is to changes in the size of the total alveolar epithelial cell population.

Indeed, letting $A_2 = A_2^+ + A_2^-$, we find that $A_1$ and $A_2$ are constant when

$$0 = K_A K_{A1}(K_{A1} - A_1)c + b\delta A_1(K_{A1} - A_1) - r_2 \sigma K_{A1}^2 A_1, \tag{2.2}$$

where $c = r_2 - \delta - \sigma > 0$ and $b = K_A \delta - r K_{A1} < 0$. Putting $x = K_{A1} - A_1$, the steady state condition is

$$h(x) = -bx^2 + K_{A1}(cK_A\delta + b\delta + r_2\sigma K_1)x - r\sigma K_{A1}^3 = 0.$$

Since $h(0) = -r\sigma K_{A1}^3 < 0$ and $h(K_{A1}) = aK_A K_{A1}^2 \delta > 0$, there exists a unique solution $\bar{x}$ so that $0 < \bar{x} < K_{A1}$. This corresponds to a unique steady state $0 < \bar{A}_1 < K_{A1}$. (The other solution corresponds to a steady state where $A_1 > K_{A1}$.) Furthermore, we find that the corresponding steady-state value of $A_2$ is $\sigma\bar{A}_1 K_{A1}/\delta(K_{A1} - \bar{A}_1) > 0$. Moreover, from the differential equation for $A_2$, we find that $\bar{A}_1 + \bar{A}_2 < K_A$.

At the steady-state solution, the Jacobian matrix is given by

$$J = \begin{bmatrix} -\sigma - \frac{\delta}{K_{A1}}\bar{A}_2 & \frac{\sigma\bar{A}_1}{\bar{A}_2} \\ \bar{A}_2\left(\frac{\delta}{K_{A1}} - \frac{r_2}{K_A}\right) & -r_2\frac{\bar{A}_2}{K_A} \end{bmatrix}, \tag{2.3}$$

so that stability of the solution is determined by the roots of

$$|J - \lambda I| = \lambda^2 + \left(\sigma + \frac{\delta}{K_{A1}}\bar{A}_2 + r\frac{\bar{A}_2}{K_A}\right)\lambda - \sigma\bar{A}_1\left(\frac{\delta}{K_{A1}} - \frac{r_2}{K_A}\right).$$

Since all coefficients of this equation in $\lambda$ are positive, we have either two negative roots or two complex roots with negative real part. In either case, the steady-state solution of interest is stable.

At steady sate, the population of macrophages is $\bar{M} = r_M/\sigma_M$.

## 2.2. Within-host model of coronavirus

### 2.2.1. Model overview

In this subsection, we propose a within-host compartmental model of the innate immune response to coronavirus infection. As before, cellular compartments are measured in units of millions of cells. In

**Table 1.** Model variables.

| variable | biological meaning |
| --- | --- |
| $A_1$ | type I alveolar cells |
| $A_2^+$ | susceptible (ACE2-positive) type II alveolar cells |
| $A_2^-$ | immune (ACE2-negative) type II alveolar cells |
| $A_2^{+*}$ | ACE2-positive type II alveolar cells that are stimulated by interferons |
| $A_2^{-*}$ | ACE2-negative type II alveolar cells that are stimulated by interferons |
| $I$ | infectious type II alveolar cells |
| $I^*$ | infectious type II alveolar cells stimulated by interferons |
| $D$ | apoptotic alveolar cells |
| $F$ | concentration of interferons |
| $X$ | concentration of chemokines |
| $M$ | inactivated innate immune cells |
| $M^*$ | activated innate immune cells |
| $T$ | concentration of toxins |
| $V$ | concentration of free virus |

addition to cells, the model with infection includes small chemical mediators and virus particles. These other compartments are measured in units of density (for example nM). This choice of units is motivated by the structure of the alveolar epithelial region, in which epithelial cells are coated in a thin layer of fluid, where smaller particles are suspended. Indeed, the alveolar epithelial cells constitute an alveolar surface with an area about 130 m² that is covered by alveolar fluid with a volume of about 40 ml [32], or about $C_1 = 20$ ml per lung. The free virus is suspended in this alveolar fluid. Interferons and chemokines are dispersed into a greater pool of fluid, which includes interstitial fluid and capillary fluid. This greater pool of lung fluid has a volume about 500 ml [32], or about $C_2 = 250$ ml per lung.

The variable names and the corresponding biological meaning are listed in table 1. As in the virus-free model, in the presence of virus, epithelial cells proliferate, differentiate, transition between ACE2-positive and ACE2-negative classes and die. In the presence of virus, cells are also subject to toxin-induced cell death, infection and interferon stimulation. Below we present a flexible model for a cell's functional response to such stimuli.

### 2.2.2. Modelling a cell's functional response to a stimulus

In the model equations, the function

$$f(C, S; K, q) = \frac{S}{S + q\dfrac{C}{2} + K} \tag{2.4}$$

determines the degree to which the population's functional response rate is saturated with respect to some stimulant. Here, $C$ represents the concentration of cells, $S$ the concentration of stimulant and $K$ represents the dissociation constant between the stimulant and the cell. In this model, the response rate is half-maximal when $S = q(C/2) + K$.

To motivate this functional form, note that the functional response of the cell is initiated by the binding of cell-surface receptors to ligands associated with the stimulant (e.g. viral proteins, cytokines and chemokines). We assume that this initial step is fast compared with the overall response time, so that $f(C, S; K, q)$ can be considered as the fraction of cells that are sufficiently stimulated to carry out the response. Alternatively, we may consider the response to be homogeneous throughout the population, in which case $f(C, S; K, q)$ represents the fraction of the maximal response rate that each cell achieves.

The concentration of stimulant required to induce a half-maximal response rate is controlled by both the dissociation constant and the cellular density, as a single stimulating particle will generally be able to stimulate at most one cell (e.g. one viral particle can infect at most one cell). Moreover, a cell may need to contact multiple stimulating particles in order to respond maximally. Hence we have introduced a

parameter $q$ to represent the number of stimulating contacts that a cell requires to trigger a maximal functional response rate. Note that when the cellular density is low, the dissociation constant between the stimulating particle and the cell is the primary determinant of the response rate. This dissociation constant is related to the dissociation constant between the stimulant's ligand and the cell-surface receptor ($K_{LR}$). In particular, we expect that $K \leq K_{LR}$ since the stimulating particle may induce multiple ligand–receptor bonds.

In cases where (i) the stimulant is not a small particle suspended in the alveolar fluid, or (ii) the responding population is present at low density, the saturation of the cell's functional response rate is instead modelled as

$$g(S; K) = \frac{S}{S + K}.$$

### 2.2.3. Model equations

In the presence of virus, alveolar epithelial cells are subject to toxin-induced cell death, infection and interferon stimulation. Indeed, activated immune cells release toxins including oxidants, proteinase-containing granuoles, and neutrophil extracellular traps (NETS) in an attempt to limit viral dissemination [33,34]. However, these toxins may also cause cell death and tissue damage. The *per capita* rate of cell death in response to toxins is modelled as

$$rg(T; K_T) := \frac{rT}{T + K_T},$$

where $r$ is the maximal response rate, and $K_T$ is the half saturation constant. (See (A.5) *Parameters describing the production and actions of toxins* for additional details). Susceptible type II cells can become infectious due to exposure to virus or become protected due to interferon stimulation. We let $\alpha$ and $\beta$ be the maximal rates at which cells transition to the interferon-protected and infectious classes, respectively, so that cells transition to the interferon-protected class at a *per capita* rate

$$\alpha f (A, F; K_F, q_F),$$

and transition to the infectious class at a *per capita* rate

$$\beta f \left( A_2^+, V; K_V, \frac{q_V}{C_1} \right).$$

Above, $A$ is the total concentration of alveolar cells (in pM) that are not yet treated by interferons. That is, $A = ((A_2^+ + A_2^- + A_1 + I)/C_1)10^{-2}/6.02$, where $10^{-2}/6.02$ is the conversion factor converting units of $10^6$ cells ml$^{-1}$ to pM. We assume that the interferon-stimulated cells, $A_2^{+*}$ and $A_2^{-*}$, lose their protected status at rate $\mu$. Moreover, interferon-protected [35] and infectious cells do not proliferate. The equations describing untreated type I and II cells are then:

$$\begin{cases} \dfrac{dA_1}{dt} = a(A_2^+ + A_2^- + A_2^{+*} + A_2^{-*}) - \sigma_A A_1 - rg(T; K_T)A_1, \\[2mm] \dfrac{dA_2^+}{dt} = r_2 \left( 1 - \dfrac{A_T}{K_A} \right) A_2^+ + a_2^- A_2^- - (a + a_2^+ + \sigma_A)A_2^+ \\[2mm] \qquad - s\alpha f (A, F; K_F, q)A_2^+ - \beta f \left( A_2^+, V; K_V, \dfrac{q_V}{C_1} \right) A_2^+ - rg(T; K_T)A_2^+ + \mu A_2^{+*}, \\[2mm] \dfrac{dA_2^-}{dt} = r_2 \left( 1 - \dfrac{A_T}{K_A} \right) A_2^- + a_2^+ A_2^+ - (a + a_2^- + \sigma_A)A_2^- \\[2mm] \qquad - \alpha f (A, F; K_F, q)A_2^- - rg(T; K_T)A_2^- + \mu A_2^{-*}, \end{cases}$$

where $A_T = A_1 + A_2^+ + A_2^- + A_2^{+*} + A_2^{-*} + I + I^*$ is the total number of alveolar type I and II cells. The equations for interferon-stimulated cells are

$$\begin{cases} \dfrac{dA_2^{+*}}{dt} = \alpha f (A, F; K_F, q)A_2^+ - (\mu + \sigma_A + a)A_2^{+*} - rg(T, K_T)A_2^{+*}, \\[2mm] \dfrac{dA_2^{-*}}{dt} = \alpha f (A, F; K_F, q)A_2^- - (\mu + \sigma_A + a)A_2^{-*} - rg(T; K_T)A_2^{-*}. \end{cases}$$

Next, we describe the dynamics of infectious type II cells. These cells are recruited by the infection of susceptible type II cells. We model four mechanisms for the removal of infectious cells: (i) natural death

at *per capita* rate $\sigma_A$; (ii) virus-induced apoptosis at *per capita* rate $\sigma_I$; (iii) toxin-induced death at *per capita* rate $rg(T; K_T)$; and (iv) interferon stimulation, the last of which yields interferon-stimulated infectious cells ($I^*$). The equations for $I$ and $I^*$ are as follows:

$$
\begin{cases}
\dfrac{\mathrm{d}I}{\mathrm{d}t} = \beta f\left(A_2^+, V; K_V, \dfrac{q_V}{C_1}\right)A_2^+ - \alpha f(A, F; K_F, q)I - (\sigma_I + \sigma_A)I - rg(T; K_T)I, \\[4mm]
\dfrac{\mathrm{d}I^*}{\mathrm{d}t} = \alpha f(A, F; K_F, q)I - (\sigma_I + \sigma_A)I^* - rg(T; K_T)I^*.
\end{cases}
$$

Apoptotic infectious cells, $D$, are recruited through infection-induced, natural and toxin-induced death of infectious cells at rates $\sigma_I (I + I^*)$, $\sigma_A (I + I^*)$ and $rg(T; K_T)(I + I^*)$, respectively. In addition, these cells are cleared by innate immune cells at a rate of $(k_{M0}(M/C_1) + k_M(M^*/C_1))D$, where $k_{M0}$ and $k_M$ are the clearance rates for activated and inactivated immune cells, respectively. Note we have divided $M$ and $M^*$ by $C_1$ as the clearance rates are measured in volume per million cells. The equation for apoptotic infectious cells is then

$$
\frac{\mathrm{d}D}{\mathrm{d}t} = (\sigma_I + \sigma_A)(I + I^*) - \left(k_{M0}\frac{M}{C_1} + k_M\frac{M^*}{C_1}\right)D + rg(T; K_T)(I + I^*).
$$

Next, we describe the dynamics of the innate immune cells. For simplicity, we group innate immune cells into activated immune cells, $M^*$, and inactive/resting immune cells, $M$. We assume resting immune cells have a natural recruitment rate $r_M$ and a natural decay rate $\sigma_M$. Inactive immune cells can be also recruited by chemokines, which occurs at rate $r_M^* g(X; K_X)$. Here, $r_M^*$ is the maximum recruitment rate of immune cells attracted by chemokines and $K_X$ is the half-saturation constant for chemokine-mediated recruitment. In our model, immune cells can be activated after engulfing virus or apoptotic cells, which occurs with a second-order rate constant $k_{M0}$. Active and resting immune cells also function to clear toxins, and this results in death. Finally, $\sigma_M^*$ is the decay rate of activated immune cells, and $\sigma_M$ is that of inactive immune cells. The equations describing innate immune cells are

$$
\begin{cases}
\dfrac{\mathrm{d}M^*}{\mathrm{d}t} = k_{M0}M\left(V + \dfrac{D}{C_1}\right) - \rho_T M^* - \sigma_M^* M^* - k_M M^* T, \\[4mm]
\dfrac{\mathrm{d}M}{\mathrm{d}t} = r_M + r_M^* g(X; K_X) - \sigma_M M - k_{M0}M\left(V + \dfrac{D}{C_1}\right) - k_{M0}MT.
\end{cases}
$$

We assume active immune cells produce toxins at a rate $\rho_T$. In addition, both active and resting immune cells clear toxins with second-order rate constants of $k_M$ and $k_{M0}$, respectively. Then the equation for toxins is

$$
\frac{\mathrm{d}T}{\mathrm{d}t} = \rho_T M^* - k_{M0}MT - k_M M^* T.
$$

We suppose that interferons are mainly produced by infectious epithelial cells and activated immune cells. However, since coronaviruses like SARS-CoV-2 have evolved mechanisms to counter the production of interferons in infectious epithelial cells [36,37], we suppose the production rate by infectious cells, $\rho_{F_2}$, is much smaller than the production rate by activated immune cells, $\rho_{F_1}$. In fact, we set the baseline value of $\rho_{F_2} = 0$ in our numerical simulations. Interferons are subject to natural decay at rate $\sigma_F$. The equation describing the interferons is

$$
\frac{\mathrm{d}F}{\mathrm{d}t} = \frac{1}{C_2}(\rho_{F_2}I + \rho_{F_2}I^* + \rho_{F_1}M^*) - \sigma_F F,
$$

where we have divided the production term by $C_2$ as the rate of production is measured as pmoles per million cells, while interferons are measured in units of density (pM).

Chemokines are produced by a variety of cells. In our model, we assume that the infectious, interferon-stimulated and activated immune cells produce chemokines at the same *per capita* rate $\rho_X$, while chemokines decay at rate $\sigma_X$. The equation describing chemokines is

$$
\frac{\mathrm{d}X}{\mathrm{d}t} = \frac{\rho_X}{C_2}(I + I^* + A_2^* + M^*) - \sigma_X X.
$$

We suppose that free virus particles in the alveolar region are produced by infectious cells at rate $\rho_V$, and interferon-stimulated infectious cells at a much smaller rate, $\rho_V^*$. We model two mechanisms for the

removal of free virus particles: (i) virus particles are engulfed by resting and active immune cells at rates $k_{M0}$ and $k_M$, respectively; (ii) virus particles decay naturally at rate $\sigma_V$. The equation describing free virus is

$$\frac{dV}{dt} = \frac{1}{C_1}(\rho_V^* I^* + \rho_V I - (k_{M0}M + k_M M^*)V) - \sigma_V V.$$

Combining the above discussions, our within-host model for the innate immune response to SARS-CoV-2 is the following system in 14 variables and 39 parameters

$$\frac{dA_1}{dt} = aA_2 - \sigma_A A_1 - rg(T; K_T)A_1,$$

$$\frac{dA_2^+}{dt} = r_2\left(1 - \frac{A_T}{K_A}\right)A_2^+ + a_2^- A_2^- - (a + a_2^+ + \sigma_A)A^+$$

$$\quad - \alpha f(A, F; K_F, q)A_2^+ - \beta f\left(A_2^+, V; K_V, \frac{q_V}{C_1}\right)A_2^+ - rg(T; K_T)A_2^+ + \mu A_2^{+*},$$

$$\frac{dA_2^-}{dt} = r_2\left(1 - \frac{A_T}{K_A}\right)A_2^- + a_2^+ A_2^+ - (a + a_2^- + \sigma_A)A_2^-$$

$$\quad - \alpha f(A, F; K_F, q)A_2^- - rg(T; K_T)A_2^- + \mu A_2^{-*},$$

$$\frac{dA_2^{+*}}{dt} = \alpha f(A, F; K_F, q)A_2^+ - (\mu + \sigma_A + a)A_2^{+*} - rg(T, K_T)A_2^{+*},$$

$$\frac{dA_2^{-*}}{dt} = \alpha f(A, F; K_F, q)A_2^- - (\mu + \sigma_A + a)A_2^{-*} - rg(T; K_T)A_2^{-*},$$

$$\frac{dI}{dt} = \beta f\left(A_2^+, V; K_V, \frac{q_V}{C_1}\right)A_2^+ - \alpha f(A, F; K_F, q)I - (\sigma_I + \sigma_A)I - rg(T; K_T)I,$$

$$\frac{dI^*}{dt} = \alpha f(A, F; K_F, q)I - (\sigma_I + \sigma_A)I^* - rg(T; K_T)I^*,$$

$$\frac{dD}{dt} = (\sigma_I + \sigma_A)(I + I^*) - \left(k_{M0}\frac{M}{C_1} + k_M\frac{M^*}{C_1}\right)D + rg(T; K_T)(I + I^*),$$

$$\frac{dF}{dt} = \frac{1}{C_2}(\rho_{F_2}I + \rho_{F_2}I^* + \rho_{F_1}M^*) - \sigma_F F,$$

$$\frac{dX}{dt} = \frac{\rho_X}{C_2}(I + I^* + A_2^* + M^*) - \sigma_X X,$$

$$\frac{dT}{dt} = \rho_T M^* - k_{M0}MT - k_M M^* T,$$

$$\frac{dM^*}{dt} = k_{M0}M\left(V + \frac{D}{C_1}\right) - \rho_T M^* - \sigma_M^* M^* - k_M M^* T,$$

$$\frac{dM}{dt} = r_M + r_M^* g(X; K_X) - \sigma_M M - k_{M0}M\left(V + \frac{D}{C_1}\right) - k_{M0}MT$$

and

$$\frac{dV}{dt} = \frac{1}{C_1}(\rho_V^* I^* + \rho_V I - (k_{M0}M + k_M M^*)V) - \sigma_V V,$$

(2.5)

where $A_2 = A_2^+ + A_2^- + A_2^{+*} + A_2^{-*}$ is the total number of uninfected type II epithelial cells.

## 2.2.4. Stability of the disease-free equilibrium

To compute the basic reproductive number and determine the stability of the disease-free equilibrium, we segregate the state variables into three classes: diseased variables $x = [I, I^*, V]$, inflamed variables $w = [D, F, X, T, M^*, A^*, A^{+*}]$ and healthy variables $y = [A_1, A_2^+, A_2^-, M]$. The disease-free equilibrium is $(\bar{x}, \bar{w}, \bar{y}) = (0, 0, \bar{y})$, where $\bar{y}$ is characterized in §2.1.2. The Jacobian matrix at the disease-free equilibrium can be decomposed as

$$\begin{bmatrix} R - S & 0 & 0 \\ * & W & 0 \\ * & * & Y \end{bmatrix},$$

(2.6)

where $R = \begin{bmatrix} 0 & 0 & \frac{2C_1\beta\bar{A}_2^+}{2C_1K_V + q_V\bar{A}_2^+} \\ 0 & 0 & 0 \\ \frac{\rho_V}{C_1} & \frac{\rho_V^*}{C_1} & 0 \end{bmatrix}$, $S = \begin{bmatrix} \sigma_I + \sigma_A & 0 & 0 \\ 0 & \sigma_I + \sigma_A & 0 \\ 0 & 0 & \frac{k_{M0}\bar{M}}{C_1} + \sigma_V \end{bmatrix}$ and

$$W = \begin{bmatrix} \frac{-K_{M0}\bar{M}}{C_1} & 0 & 0 & 0 & 0 & 0 & 0 \\ 0 & -\sigma_F & 0 & 0 & \frac{\rho_{F_1}}{C_2} & 0 & 0 \\ 0 & 0 & -\sigma_X & 0 & \frac{\rho_{F_1}}{C_2} & \frac{\rho_{F_1}}{C_2} & \frac{\rho_{F_1}}{C_2} \\ 0 & 0 & 0 & -\frac{k_{M0}\bar{M}}{C_1} & \rho_T & 0 & 0 \\ \frac{K_{M0}\bar{M}}{C_1} & 0 & 0 & 0 & -(\sigma_M + \rho_T) & 0 & 0 \\ 0 & \alpha\frac{\partial f}{\partial F}(A, F; K_F, q)\bar{A}_2^+ & 0 & 0 & 0 & -(\mu + \sigma_A + a) & 0 \\ 0 & \alpha\frac{\partial f}{\partial F}(A, F; K_F, q)\bar{A}_2^- & 0 & 0 & 0 & 0 & -(\mu + \sigma_A + a) \end{bmatrix}.$$

Clearly, $W$ has only negative eigenvalues and from (§2.1.2), the eigenvalues of $Y$ also have negative real part. Thus, the disease-free equilibrium is stable if the eigenvalues of $R - S$ have negative real part. From [38], the eigenvalues of $R - S$ have negative real part if and only if the spectral radius of the next generation matrix, $RS^{-1}$, is less than one. In this case, that radius, which also represents the model's basic reproductive number is

$$R_0 = \sqrt{\frac{2C_1\beta\bar{A}_2^+}{(2C_1K_V + q_V\bar{A}_2^+)(k_{M0}\bar{M} + \sigma_V C_1)}\frac{\rho_V}{(\sigma_I + \sigma_A)}}. \tag{2.7}$$

Note that $R_0$ is unitless. In biological terms, $R_0$ represents the greatest factor by which an initial vector $x_0$ will be amplified after one generation. It is the geometric mean of the infectious cells generated by each pfu ml$^{-1}$ of virus after one generation

$$\frac{2C_1^2\beta\bar{A}_2^+}{(2C_1K_V + q_V\bar{A}_2^+)(k_{M0}\bar{M} + \sigma_V C_1)}, \tag{2.8}$$

and the concentration of virus generated by each infectious cell after one generation

$$\frac{\rho_V}{C_1(\sigma_I + \sigma_A)}. \tag{2.9}$$

From the expression for $R_0$, we can see how individual mechanisms contribute to resistance to infection. Resistance can be achieved by (i) augmenting the rates of viral decay and phagocytosis by resting resident alveolar macrophages, (ii) decreasing the number of susceptible cells, the affinity of virus for cells, or the number of ACE2 receptors per cell, (iii) reducing the rate of viral production by infectious cells, and (iv) reducing the lifespan of infectious cells. We note that the rates of viral decay and phagocytosis are impacted by the body's innate immune surveillance system. Hence this system has the capacity to endow resistance. On the other hand, we see that the innate immune response, which is represented by the variables associated with inflammation, does not influence the stability of the disease-free equilibrium, as the associated variables and parameters do not impact the model's basic reproductive number. Indeed, since the eigenvalues of the matrix associated with inflammation ($W$) are negative, the innate immune response does not alter the stability of the disease-free equilibrium. Importantly, this means that the model does not exhibit runaway inflammation. It does not, however, mean that the innate immune response is unimportant or ineffective. Indeed, while unable to alter the stability of the disease-free equilibrium, this response can dramatically alter the course of infection and/or severity of the endemic equilibrium, by reducing numbers of susceptible cells (interferon stimulation), reducing viral production rates (interferon stimulation), and increasing rates of phagocytosis (chemokine-induced recruitment of innate immune cells).

When parametrized as described in §3.1 and appendix A, we find $R_0 = 2.85$, which is similar to the estimate from [10]. In addition, by evaluating (2.8) and (2.9), we estimate the ability of free virus and susceptible cells to propagate the infection. We find that each pfu ml$^{-1}$ of free virus produces on average just 0.72 infectious cells, while each infectious cell produces on average 11.18 (pfu ml$^{-1}$) of virus. Alternatively, on average, each pfu of virus produces only 0.036 infectious cells. Although our ordinary differential equation model can only describe the average/deterministic course of infection, the limited ability of virus to produce infectious cells suggests that when the initial viral load is low, stochastic effects may prevent the infection from taking root. Hence the model supports the efficacy of preventative measures aimed at reducing the quantity of virus to which an individual is exposed.

Finally, it is interesting to consider how individual variability can impact the stability of the disease-free equilibrium, that is, the host's susceptibility/resistance to infection. Potential sources of variable susceptibility include variability in the susceptibility of alveolar type II cells to infection, variability of

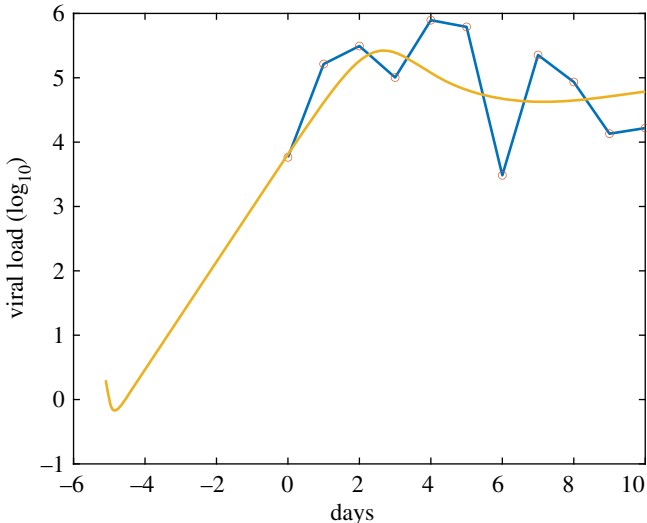

**Figure 1.** Model fit to viral load data. The x-axis is the number of days since the onset of symptoms, and the y-axis is the saliva viral load (unit is $\log_{10}$ pfu/ml). The dots correspond to the average saliva viral load data of 23 patients in [47]. The curve is the simulated saliva viral load, where it is assumed that the viral load in the alveolar region is two magnitudes larger than that in the saliva.

alveolar macrophage function and variability in lung surfactant. It is uncertain if there exists significant individual variability in the susceptibility of alveolar type II cells to infection. However, such variability could result from variable ACE2 expression, if present in a population [39–42]. In our model, changes in ACE2 expression can impact $R_0$, and hence the stability of the disease-free susceptibility of the host, by altering the number of susceptible cells in the lung ($A_2^+$) or the number of ACE2 receptors per cell ($q_V$). Meanwhile, chronic disease and ageing are associated with macrophage defects, including defective clearance of pathogens [43–45]. In our model, such defects increase the host's susceptibility by reducing $k_{M0}$. Finally, changes in the expression of surfactant proteins [46] could alter host susceptibility through $\rho_V$, which represents the rate of virus decay in the alveolar region. The presence of SARS-CoV-2 antigens, due to vaccination or previous infection, would also reduce susceptibility through $\rho_V$.

# 3. Numerical simulations

In this section, we present numerical simulations of the model in Matlab: we estimate a few of the most uncertain parameters, report model predictions of the innate immune response, and study the impact of host/pathogen variability and interventions/prophylactics on the course of the infection.

## 3.1. Model fitting

Here, we fit a small number of the model parameters, namely $\gamma$, $\rho_V$ and $T_0$, to viral load data using a least-squared error scheme. The data for fitting was sourced from [47], where the saliva viral load of 23 patients was monitored and recorded daily, beginning at the onset of the symptoms. As our model tracks viral load in the *alveolar fluid* rather than the saliva, we increase the viral loads in [47] by two orders of magnitude prior to fitting, to reflect anticipated differences in the viral loads in these two fluids. In particular, we assume the alveolar viral load is much greater than the saliva viral load as the saliva is replenished much more quickly than the alveolar fluid. We assume that the infection in the alveolar region is initiated by a small dose of free virus.

In figure 1, we plot the simulated alveolar viral load data, starting from the initial infections up until 10 days after the onset of symptoms, alongside the scaled empirical data. Our simulations show that the model provides a reasonable fit of the empirical data. Although the initial dip in the viral load may seem strange, it is readily explained: this dip occurs because there is a delay of several hours before infected cells begin to produce new virus. During this time, the concentration of virus falls monotonically due to thermal inactivation. Moreover, in the absence of adaptive immunity, we predict the viral load will undergo biphasic decay leading to chronic infection [48].

All model parameter values (both fit and estimated) can be found in table 2, along with references and estimated biological range. Initial values are listed in table 3 (the initial values not listed in the table are zeros).

## 3.2. Dynamics and efficacy of the innate immune response

Next, we examine the model-predicted response to the infection, including the dynamics of the epithelial cell population (figures 8–10), the levels of cytokines, chemokines and toxins (figure 11), and the infiltration and activation of innate immune cells (figure 12). We see that a fraction of alveolar epithelial cells are lost to the infection, this includes a small loss of type I alveolar cells, probably due to exposure to extracellular toxins. Meanwhile, markers of inflammation, including interferons, chemokines, toxins and activated innate immune cells show a sharp increase, similar to viral load, before falling to low levels approximately three weeks post infection. This innate immune response appears sufficient to stymie, but not eliminate the infection, as is evidenced by the persistence of infectious cells, virus and markers of inflammation. The endemic equilibrium is characterized by an increase in the total number of innate immune cells patrolling the alveolar region, a fraction of which remain active. Also, while the total number of alveolar type I cells partially recovers, it remains beneath the disease-free equilibrium value. In summary, the model predicts that the innate immune response controls, but does not eliminate the infection, so that the initial acute phase of infection gives way to a sustained chronic phase of infection.

## 3.3. Interventions and prophylactics

### 3.3.1. The impact of interferon treatment on the course of infection

Since SARS-CoV-2 has multiple mechanisms to counter the production of interferons [36,37], and interferon-stimulated infectious cells produce significantly less virus than untreated ones, interferons have been proposed as a possible drug therapy for the disease [89,90]. In this set of simulations, we fix the parameter values from the first simulation but increase the concentration of type I interferons in the alveolar region by a fixed amount. We consider two treatment schedules, where treatment is adopted at day 1 or 3, and two treatment levels, where the concentration of type I interferons is increased by 10% or 20% of the half saturation constant for the antiviral interferon response ($K_F$). Looking at figure 2, we see that all treatments can effectively reduce the viral load; however, the high-dose treatments ($\Delta F = 0.2\ K_F$) are more effective at reducing viral loads than lower-dose treatments ($\Delta F = 0.1\ K_F$). Although the timing of treatment initiation does not significantly impact the ultimate reduction in the viral load, early treatment has the potential to limit the size of the infection (figure not shown). Unfortunately, interferon treatment also limits the ability of the alveolar epithelium to heal (see figure 2). As a result, it may be necessary to limit the duration of interferon treatment. In short, while interferon treatment can be beneficial early in the infection, it may be detrimental at latter stages of infection when the lung is attempting to regenerate. Because such limited treatment may be insufficient to eliminate the infection, our results suggest that, at least in that absence of additional adaptive immune responses, interferon treatment is not a feasible stand-alone therapy. Instead, interferon treatment may offer a means of potentiating the effects of other therapies.

### 3.3.2. The impact of antiviral treatment on the course of infection

Multiple antiviral drugs are in clinical trials for the treatment of SARS-CoV-2 [91]. In [92], the authors studied the *in vitro* antiviral activity of chloroquine and hydroxychloroquine for SARS-CoV-2 and found the *in vitro* antiviral $EC_{50}$ of these drugs to be 5.47 and 0.72 µM, respectively. In [93], the authors tested the effectiveness of seven drugs and found the *in vitro* $EC_{50}$ of remdesivir and chloroquine to be 1.13 and 0.77 µM, respectively. The lung pharmacokinetics in [94] show that different remdesivir doses lead to drug concentrations between 0.5 and 5 µM, with a drug half-life about 3 h. Clinical trials showed that patients treated with hydroxychloroquine did not have lower death rates than those who received the standard care [95] while remdesivir treatment was found to have limited benefit [93,96]. In this set of simulations, we fix the parameter values from the first simulation, except the production rate of virus is reduced to represent the impact of antiviral drug therapy. In particular, we simulate the effectiveness of antiviral drug therapies which can reduce the viral production rate by 50% or 90%. For simplicity, we assume that the effect of antiviral drug

**Table 2.** Parameter values used in the simulations.

| parameter | symbol | value | biological range | ref. |
|---|---|---|---|---|
| natural death rate for $A_1$, $A_2$ cells | $\sigma_A$ | $0.00035$ h$^{-1}$ | | [49] |
| natural death rate for $M$ cells | $\sigma_M$ | $0.0005$ h$^{-1}$ . | [0.0001, 0.0005] | [50,51] |
| natural death rate for $M^*$ cells | $\sigma_M^*$ | $0.02$ h$^{-1}$ | [0.008, 0.05] | [52,53] |
| growth rate of $A_2$ cells | $r_2$ | $0.055$ h$^{-1}$ | | [54] |
| rate at which ACE2-expression of the population equilibrates | $\gamma$ | 7.73 | | fit |
| maximal differentiation rate of $A_2$ cells into $A_1$ cells | $\delta$ | $0.006$ h$^{-1}$ | | [55] |
| parameter controlling differentiation of $A_2$ cells | $K_{A1}$ | $20.32 \times 10^4$ $10^6$ cells | | estimated |
| fraction of $A_2$ cells susceptible in the healthy lung | $p_+$ | 0.05 | [0.01, 0.3] | [24,56,57] |
| rate at which cells transition to the immune class | $a_2^+$ | $5 \times 10^{-4}$ h$^{-1}$ | | estimated, [24,56] |
| rate at which cells transition to the susceptible class | $a_2^-$ | $2.8 \times 10^{-5}$ h$^{-1}$ | | estimated, [24,56] |
| carrying capacity of $A_1$, $A_2$ cells | $K_A$ | $5.3 \times 10^4$ $10^6$cells | | [20,21,54] |
| recruitment rate of $M$ cells | $r_M$ | 3 ($10^6$ cells h$^{-1}$) | [0.6, 3] | [50,51,58] |
| recruitment rate of $M^*$ cells | $r_M^*$ | 350 ($10^6$ cells h$^{-1}$) . | [100, 400] | [58,59] |
| maximal effective contact rate between $V$ and $A_2$ | $\beta$ | $\frac{1}{6}$ h$^{-1}$ | [1/10, 1/3] | [60,61] |
| $K$ for saturation of infection rate | $K_V$ | $10^3$ ($10^6$ pfu/ml) | $[10^3, 10^6]$ ($10^6$ pfu/ml) | estimated |
| $q$ for saturation of infection rate | $q_V$ | 1 | | estimated |
| decay rate of $V$ | $\sigma_V$ | $\frac{1}{3}$ h$^{-1}$ | [1/10, 1/3] | [62–64] |
| production rate of $V$ | $\rho_V$ | 3.18 ($10^6$ pfu/$10^6$ cell h) | [1, 100] | fit, [64,65] |
| production rate of $V$ for interferon-treated cells | $\rho_V^*$ | $3.18 \times 10^{-2}$ ($10^6$ pfu/$10^6$ cells h) | $[1, 100] \times 10^{-3}$ | [66] |
| rate at which interferon-stimulation decays | $\mu$ | $0.005$ h$^{-1}$ | [0.004, 0.006] | [67] |
| rate of apoptosis for infectious cells | $\sigma_I$ | $\frac{1}{72}$ h$^{-1}$ | | [68] |
| maximal rate of phagocytosis for resting immune cells | $k_{M0}$ | $10^{-4}$ (ml/$10^6$cells h) | $[0.5, 1] \times 10^{-4}$ | [32,58,69] |
| maximal rate of phagocytosis for active immune cells | $k_M$ | $3 \times 10^{-4}$ (ml/$10^6$cells h) | $[3, 10] \times 10^{-4}$ | [32,58,69] |

(Continued.)

**Table 2.** (*Continued.*)

| parameter | symbol | value | biological range | ref. |
|---|---|---|---|---|
| production rate of X | $\rho_X$ | 0.006 (pmol/$10^6$cells h) | [0.0006, 0.6] | [70] |
| concentration at which X induces half-maximal chemotaxis | $K_X$ | 500 pM | | [71–73] |
| production rate of F by innate immune cells | $\rho_{F_1}$ | 0.01 (pmol/$10^6$ cell h) | [0, 0.1] | [74] |
| production rate of F by epithelial cells | $\rho_{F_2}$ | 0 (pmol/$10^6$ cell h) | [0, 0.142] | [75,76] |
| decay rate of F | $\sigma_F$ | 0.35 h$^{-1}$ | [0.014, 5.20] | [77–79] |
| maximal rate of transition to antiviral state | $\alpha$ | 0.6 h$^{-1}$ | | [76] |
| K for saturation of antiviral response | $K_F$ | 100 pM | | estimated |
| q for saturation of antiviral response | $q_F$ | 40 | | estimated |
| production rate of T | $\rho_T$ | 0.12 h$^{-1}$ | [0.12, 0.23] | [80,81] |
| maximal rate of death rate due to exposure to T | $r$ | 0.1 h$^{-1}$ | [0.03, 0.07] | [82,83] |
| concentration at which T induces half-maximal cell death | $K_T$ | $3 \times 10^2$ $10^6$ NETs | | estimated |
| decay rate of X | $\sigma_X$ | 1 h$^{-1}$ | [0.5, 2] | [84–86] |
| decay rate of T | $\sigma_T$ | 0.29 h$^{-1}$ | [0.0029, 0.29] | [87,88] |
| volume of alveolar fluid (per lung) | $\varsigma_1$ | ≈20 ml | | [32] |
| total volume of fluid (per lung) | $\varsigma_2$ | ≈0.250 l | [0.200, 0.300] | [32] |
| incubation period | $T_0$ | 5 days | [2, 7] | fit |

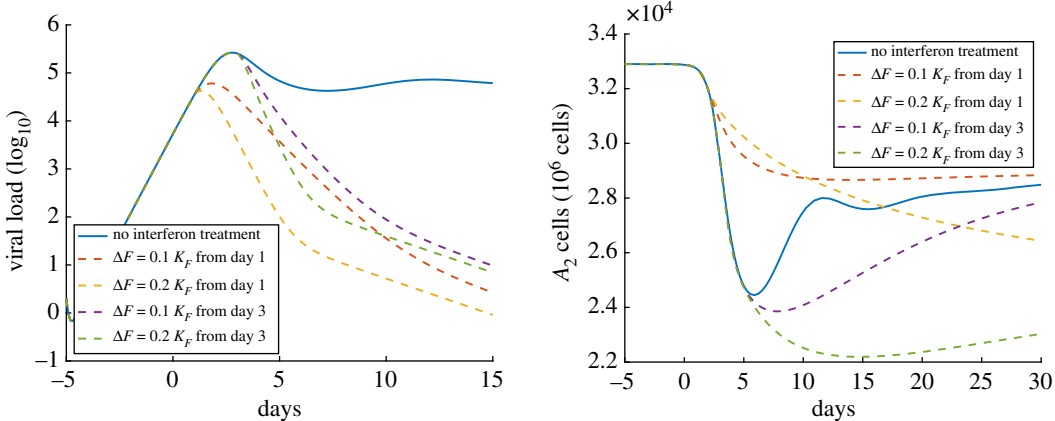

**Figure 2.** The impact of interferon treatment on the course of infection. The x-axis is the number of days since the onset of symptoms. The y-axis on the left is the viral load (unit is $\log_{10}$ pfu ml$^{-1}$), and the y-axis on the right is the number of healthy type II alveolar epithelial cells ($A_2^+ + A_2^- + A_2^{+*} + A_2^{-*}$). The solid curves on the left and right show the viral load and the number of $A_2$ cells from the first simulation with no treatment, respectively; the dashed curves on the left and right show the viral load and the number of $A_2$ cells with four different interferon treatment strategies, respectively.

**Table 3.** Baseline initial values.

| parameter | symbol | value |
| --- | --- | --- |
| type I alveolar cells | $A_1$ | $1.96 \times 10^{10}$ cells |
| type II alveolar cells expressing ACE2 | $A_2^+$ | $5\% \times 3.29 \times 10^{10}$ cells |
| type II alveolar cells not expressing ACE2 | $A_2^-$ | $95\% \times 3.29 \times 10^{10}$ cells |
| inactivated immune cells | $M$ | $5.99 \times 10^9$ cells |
| concentration of free virus | $V$ | 200 pfu ml$^{-1}$ |

therapy is constant and immediate. We also simulate two treatment schedules, where treatment is initiated at day 1 or 3.

In figure 3, we see that a 90% reduction in the rate of viral production effectively reduces the viral load in the alveolar region, while a 50% reduction in the rate of viral production has little impact on the long-term viral load, regardless of treatment timing. All treatments improve the ability of the alveolar epithelium to heal. However, while treatment that reduces viral production rates by 90% largely eliminates the infection and enables epithelial type II cells to return to pre-infection levels, antiviral treatment leading to a 50% reduction in viral production is of limited benefit. Treatment timing does not significantly impact the ultimate effectiveness of antiviral treatment; however, earlier treatment has greater potential to limit the number of cells infected and reduce the severity of tissue damage.

### 3.3.3. The impact of initial viral load on the course of infection

Preventative measures, including face masks and hand washing, can reduce the initial viral load. In the previous simulations, we took the initial concentration of the virus to be 200 pfu ml$^{-1}$. From figure 4, we see that when the initial viral load is increased by one order of magnitude the time it takes for the viral load to peak decreases by 1–2 days. However, the initial viral concentration has no impact on the peak viral load. Hence, in our model, the initial viral load predominantly impacts the time at which peak viral load is reached. We reiterate, however, that very low initial viral loads can be subject to stochastic effects that are not captured by this model.

## 3.4. Pathogen and host variability

### 3.4.1. The impact of ACE2 expression on the course of infection

Next, we study the impact of the percentage of ACE2$^+$ cells on the course of the infection. On average, we assumed that 5% of alveolar type II cells are ACE$^+$ and thereby susceptible to virus infection. From

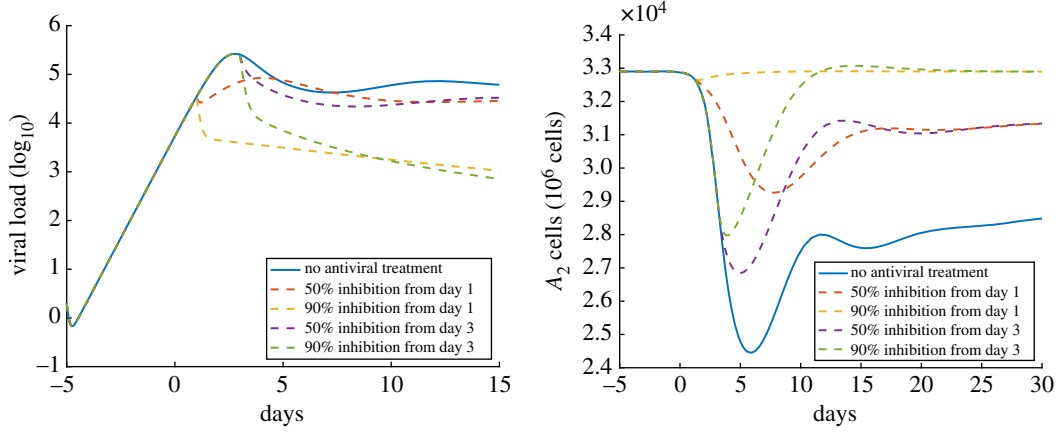

**Figure 3.** The impact of antiviral treatment on the course of infection. The x-axis is the number of days since the onset of symptoms. The y-axis on the left is the viral load (unit is $\log_{10}$ pfu ml$^{-1}$), and the y-axis on the right is the number of healthy type II alveolar epithelial cells ($A_2^+ + A_2^- + A_2^{+*} + A_2^{-*}$). The solid curves on the left and right show the viral load and the number of $A_2$ cells from the first simulation with no treatment, respectively; the dashed curves on the left and right show the viral load and the number of $A_2$ cells with four different antiviral drug therapy strategies.

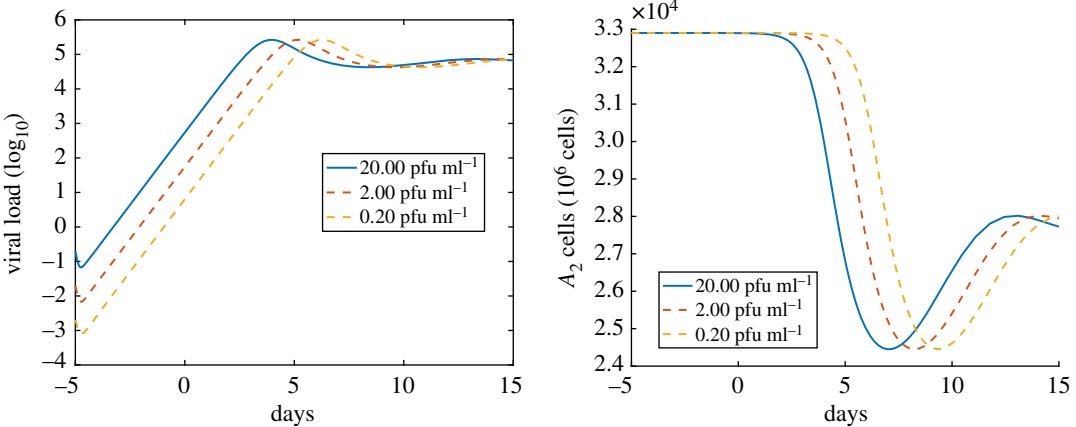

**Figure 4.** The impact of initial viral load on the course of infection. The x-axis is the number of days since the onset of symptoms. The y-axis on the left is the saliva viral load (unit is $\log_{10}$ pfu ml$^{-1}$). The y-axis on the right is the number of healthy type II alveolar epithelial cells ($A_2^+ + A_2^- + A_2^{+*} + A_2^{-*}$). The solid curves on the left and right denote the simulation with the baseline initial viral load.

figure 5, we can see that if instead 10% of the type II cells are ACE$^+$, the viral load is about one order of magnitude larger and peaks about 5 days earlier. On the contrary, if only 2.5% of cells are ACE$^+$, then it takes much longer for the viral load to peak. Finally, if only 0.5% of cells are ACE$^+$, the infection does not take hold. Hence, in addition to potentially preventing SARS-CoV-2 infection by reducing the value of $R_0$ below one, the percentage of cells that are susceptible to infection has a significant impact on the course of the illness.

### 3.4.2. The impact of epithelial-cell interferon production on the course of infection

Since SARS-CoV-2 has evolved mechanisms for inhibiting interferon (IFN) production, and some research suggests that select immune cells are specially equipped to produce type I interferons [74], our baseline simulation neglected epithelial-cell IFN production. However, since SARS-CoV-infected alveolar epithelial cells were found to produce type I IFN mRNA *in vitro*, and individual differences in IFN signalling may partially explain variability in the severity of COVID-19 symptoms [97–99], it is interesting to consider how epithelial-cell IFN production may impact the course of infection. Figure 6 shows that epithelial-cell IFN production results in a significantly faster IFN response and a marginal increase in maximal IFN levels. Moreover, epithelial-cell IFN production appears more effective at

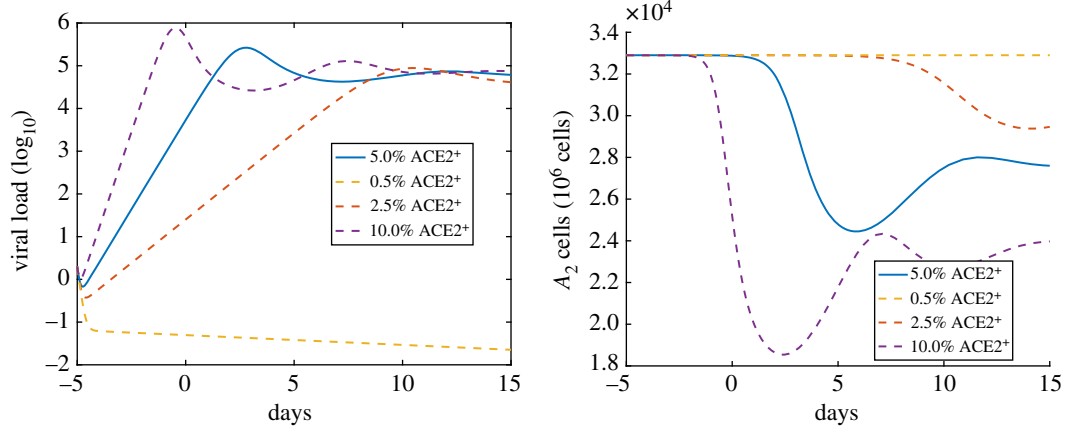

**Figure 5.** The impact of ACE2 expression on the course of infection. The x-axis is the number of days since the onset of symptoms. The y-axis on the left is the saliva viral load (unit is $\log_{10}$ pfu ml$^{-1}$). The y-axis on the right is the number of healthy type II alveolar epithelial cells ($A_2^+ + A_2^- + A_2^{+*} + A_2^{-*}$). The solid curves on the left and right denote the simulation with the baseline ACE2 expression.

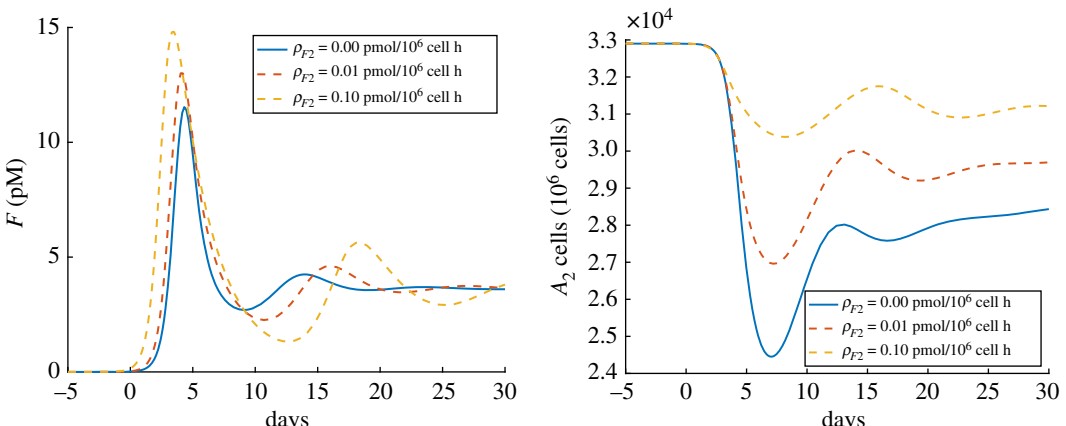

**Figure 6.** The impact of epithelial cell interferon production on the course of infection. The x-axis is the number of days since the onset of symptoms. The y-axis on the left is the concentration of type I interferons in the alveolar region. The y-axis on the right is the number of healthy type II alveolar epithelial cells ($A_2^+ + A_2^- + A_2^{+*} + A_2^{-*}$). The solid curves on the left and right denote the simulation with no epithelial cell interferon production.

preserving the population of alveolar type II cells than IFN treatment. Thus, our simulations support the idea that defective or delayed IFN signalling worsens prognosis.

### 3.4.3. The impact of spike protein:ACE2 binding affinity on the course of infection

Although it is difficult to precisely quantify the affinity of the full-length spike protein for the ACE2 receptor *in vivo*, research suggests this quantity may vary significantly (almost fivefold) between the B.1.351 SARS-CoV-2 strain and the strain first identified in Wuhan [100]. This change in affinity is driven by mutations within the spike protein. These include the substitutions D614G, found in variants Alpha, Beta, Gamma and Delta, and N501Y, found in Alpha, Beta and Gamma. Both of which have been shown to enhance binding of the spike protein to ACE2 *in vitro* and in animal models [101,102]. Hence, we investigate how the affinity of the SARS-CoV-2 spike protein for the ACE2 receptor shapes the course of the infection. Figure 7 shows that a fivefold increase in affinity results in a faster more destructive infection. Specifically, with a fivefold increase in affinity, tissue damage occurs several days earlier, and healthy alveolar type II cell numbers are approximately 40 or 20% lower in the acute and chronic phases of infection, respectively. Meanwhile, a twofold decrease in the spike protein:ACE2 binding affinity results in a relatively slow-moving, mild infection, with

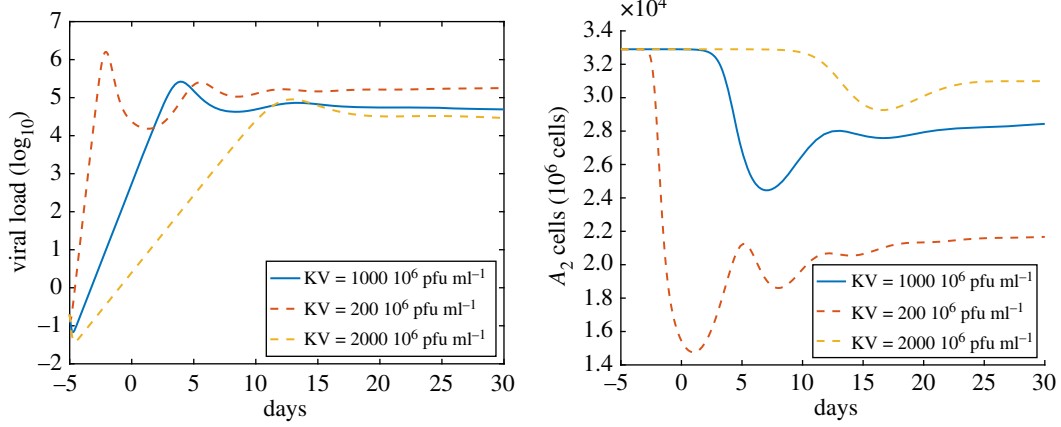

**Figure 7.** The impact of the binding affinity between ACE2 and the SARS-CoV-2 spike protein on the course of infection. The y-axis on the left is the saliva viral load (unit is $\log_{10}$ pfu ml$^{-1}$). The y-axis on the right is the number of healthy type II alveolar epithelial cells $(A_2^+ + A_2^- + A_2^{+*} + A_2^{-*})$. The solid curves on the left and right denote the baseline affinity.

limited cell damage. Interestingly, in the chronic phase of infection, the viral load is very similar for all three affinities tested. Similar to our result, previous research on a SARS-CoV-2 strain carrying a D614G spike protein mutation found this strain is more efficient at infecting some human cell lines; however, this increased infectivity did not translate to higher viral loads in the hamster lung [101]. In summary, our results support the idea SARS-CoV-2 strains with enhanced spike protein:ACE2 binding affinities have the potential to produce more severe lung pathologies irrespective of any other genetic changes they might carry.

# 4. Summary

Here, we have developed a differential equation model to study the innate immune response to SARS-CoV-2 within the alveolar epithelium. We have considered numerous variables that are probably critical determinants of the viral dynamics and the host response, including type I and type II alveolar epithelial cells, interferons, chemokines, toxins and innate immune cells. Important factors and mechanisms, including the percentage of ACE2$^+$ cells, differentiation and proliferation of alveolar epithelial cells, and tissue damage by toxins, are also described. We have characterized the model steady states and basic reproductive number, estimated most model parameters from the biological literature, and compared the model output with human viral load data. Below we summarize our major contributions and conclusions.

Our model of the alveolar epithelium includes type I and type II alveolar epithelial cells, is parametrized from available biological literature, and accounts for the proliferation and differentiation of alveolar type II cells. We find that the model has a positive steady state which is stable provided that: (i) proliferation occurs on a faster time scale than differentiation and (ii) proliferation is more sensitive to cell death than differentiation. In the future, this model can provide a useful basis for mathematical investigations of the alveolar epithelium in health and disease.

We have developed and parametrized a model of the innate immune response to SARS-CoV-2 infection in the alveolar epithelium and characterized the model's basic reproductive number/stability of the disease-free equilibrium. We estimate the reproductive number to be 2.85, which is similar to that estimated elsewhere [10]. Importantly, we have further decomposed the reproductive number in order to characterize the average number of infectious cells produced by one viral pfu and the average number of viral pfu produced by one infectious cell. We find a single viral pfu produces on average much less than one infectious cell. In fact, just 0.036 infectious cells. Hence, low initial viral loads are unlikely to initiate infection in the presence of stochasticity.

Fitting the model to viral load data, we then simulated the infection over an extended period of time to characterize the dynamics and efficiency of the innate immune response. We found that the model converges to a positive endemic equilibrium, which means that, as parametrized, the innate immune response is insufficient to clear the infection. The endemic equilibrium is characterized by partial recovery of the alveolar epithelium, low numbers of infectious cells, and persistent inflammation.

We also studied the impact of interventions and prophylactics on the course of the infection. Our simulations show that interferon therapy (at both dosage levels considered) significantly reduces the viral load, although the higher dose has greater effect. However, interferon therapy also limits the ability of the alveolar epithelium to heal. Our results suggest that interferon treatment is best administered at low doses and over short intervals of time in combination with other therapies. By contrast, our simulations show that antiviral treatment, while less effective at reducing viral load, can limit the extent of tissue damage. In particular, antiviral treatment that reduces the rate of viral production by 90% effectively controls the infection and enables epithelial cell numbers to return to pre-infection levels. However, treatment that results in a 50% reduction in the rate of viral production has only limited benefit. By contrast, our model predicts that reducing the initial viral load, for example through prophylactic measures like mask usage and hand washing, only impacts the duration of the asymptomatic period. However, this prediction is subject to the caveat that very small initial viral loads are probably subject to stochastic effects, which are not described in this model.

Finally, we studied the impact of host and pathogen variability on the course of the infection. Simulations showed that, in addition to impacting the stability of the disease-free equilibrium and basic reproductive number, the percentage of ACE2$^+$ cells has a dramatic impact on the course of infection, with higher percentages resulting in earlier and larger peak viral loads coupled with increased cell death. Variations in the rate of production of type I interferons by alveolar epithelial cells and the affinity of the SARS-CoV-2 spike protein for the ACE2 receptor had similarly dramatic effects on the course of infection. In summary, the model predicts that individual differences between patients and viral strains can significantly alter the prognosis.

Although our model incorporates many important features of SARS-CoV-2 infection within the alveolar epithelium and is based on current biological knowledge, it is also limited. Indeed, investigations into the within-host mechanisms of SARS-CoV-2 are ongoing and can inform future improvements to the model. For example, the precise function of ACE2 signalling and interferon signalling for proliferation, infection and healing, including potential cross-talk between signalling pathways, remains to be elucidated. For the time being, our model only considers ACE2 expression as a determinant of susceptibility; however, it may also be important for healing [103]. In addition, known features of the system have been neglected or simplified in constructing this model. For example, we did not explicitly model the antiviral activities of the complement system, or the effect of fever, which could both lead to dynamic changes in the rate of viral decay. Instead, we have focused on the dynamic regulation of viral load by innate immune cells. Similarly, the model is only intended to describe the innate immune response; the adaptive immune response is neglected. And, as with most mathematical models of biological systems, the parameters are uncertain. Indeed, parameter estimates are based on a variety of publications and model systems, and frequently require extrapolation to the human alveolar epithelium. Also, when modelling the impact of drug therapy, there are probably additional side effects that are not represented in the model. Moreover, it is important to acknowledge that parameter values may vary significantly from individual to individual. For example, the total number of cells, the percentage of ACE2 positive cells, and the production and clearance rates of interferons and virus may be very different for different individuals [104]. Finally, the model is limited by its medium, as differential equations necessarily portray an average view of reality that does not account for discrete quantities or chance.

Despite these limitations, we have assimilated a considerable amount of current biological knowledge into a mathematical model that can be useful for assessing our understanding of SARS-CoV-2 infection, generating predictions regarding the efficacy of treatments, and identifying factors that influence the probability and severity of SARS-CoV-2 infection.

Data accessibility. Data and relevant code for this research work are stored in GitHub: https://github.com/rnleander/Within-host_SARS-CoV-2.git and have been archived within the Zenodo repository: 10.5281/zenodo.511043.

Authors' contributions. R.N.L. developed and parametrized the model, characterized the model steady states and basic reproductive number, assisted with numerical simulations and the generation of figures, and wrote and edited the paper. Y.W. developed and parametrized the model, characterized the model steady states and basic reproductive number, procured data for model fitting, wrote the code to numerically simulate the model and generate figures, and wrote and edited the paper. W.D. helped to develop and parametrize the model, verified the model basic reproductive number and edited the paper. D.E.N. helped develop and parametrize the model and helped to write and edit the paper. Z.S. helped to develop and parametrize the model, helped with numerical simulations and edited the paper.

Competing interests. We declare we have no competing interests.

Funding. No funding has been received for this article.

# Appendix A. Model details and parametrization

## A.1. Parameters describing the demographics of alveolar epithelial cells

Alveolar epithelial cells are long-lived. The lifespan of the alveolar epithelial type I cell is reported to be 120 days [49]. We were unable to find estimates of the lifespan of the alveolar type II cell. However, as the average half-life of a ciliated lung epithelial cell is 17 months [105], we expect alveolar epithelial type II cells to be long-lived. For simplicity, we suppose type I and type II cells have equal lifespans and choose $\sigma_A = 1/120\,\mathrm{d}^{-1} = 0.00035\,\mathrm{h}^{-1}$.

According to [58], a single human lung contains on average $\bar{A}_1 = 19\,600 \pm 9000 \times 10^6$, $\bar{A}_2 = 32\,900 \pm 13\,600 \times 10^6$ and $\bar{M} = 5990 \pm 1900 \times 10^6$ type I cells, type II cells and alveolar macrophages, respectively; the total number of cells in the alveolar region of a lung is then $184\,000 \pm 65\,000 \times 10^6$. Alveolar type II cells replenish the lung by proliferating and differentiating into type I cells [55,106]. We estimate the *per capita* rate at which alveolar type II cells differentiate into alveolar type I cells at steady state as

$$a = \frac{\sigma_A \bar{A}_1}{\bar{A}_2} = 0.00021\,\mathrm{h}^{-1}.$$

The rate of differentiation, $a$ (as well as the rate of proliferation, $r_2$) should increase in response to injury in order to maintain oxygen exchange. Hence we model the rate of differentiation as

$$a = \delta\left(1 - \frac{\bar{A}_1}{K_{A1}}\right),$$

where $\delta$ is the maximal average rate of proliferation and $K_{A1}$ determines the fraction of type II cells that actively differentiate, given the current number of type I cells. The rate of differentiation in the rat lung post lipopolysaccharides (LPS)-induced injury was quantified in [55]. Here differentiated cells were observed 7 days post injury. Hence, we take

$$\delta = \frac{1}{7*24}\,\mathrm{h}^{-1} = 0.006\,\mathrm{h}^{-1}.$$

$K_{A1}$ can then be estimated as

$$K_{A1} = \frac{\bar{A}_1}{1 - (a/\delta)} = 20.32 \times 10^4\,10^6 \mathrm{cells}.$$

Note this implies approximately 3.5% of type I cells are in the process of differentiating into $A_1$ cells at steady state. While this number seems high given data from [55], it accurately captures the steady-state population structure.

Evidence suggests that alveolar type II cells are not uniformly susceptible to coronavirus infection [57]. In particular, only a fraction of these cells express receptors that enable infection by SARS-CoV-2 [30]. Hence, we let $A_2^+$ denote type II cells that are susceptible to infection and $A_2^-$ denote cells that are immune to infection. We assume cells transition from the immune to susceptible class through increased expression of cell surface receptors such as ACE2 at a rate $a_2^-$ and move from the susceptible to immune class through decreased expression of these receptors at a rate $a_2^+$, where

$$a_2^- = \gamma p_+ (a + \sigma_A)$$

and

$$a_2^+ = \gamma (1 - p_+)(a + \sigma_A).$$

It follows that when the structure of the $A_2$ cell population equilibrates a fraction, $p_+$, of these cells will be susceptible to SARS-CoV-2 infection.

Estimates of the fraction of alveolar type II cells that express ACE2 are typically low, ranging from 1 to 7% [24,56]. Hence, we take $p_+ = 0.05$. However, an *in vitro* study of SARS-CoV-1 infection of type II cells suggests that ACE2 expression and susceptibility are highly variable *in vitro*. In particular, the fraction of cells infected with SARS-CoV-1 in this study, which provides a lower bound for the susceptible fraction, varied between approximately 3 and 34%. Hence we take $\in [0.01, 0.3]$ as the range of physiological values for $p_+$. The factor $\gamma$ determines the rate at which the ACE2-expression structure of the

population equilibrates relative to the lifespan of a type II cell. We fit $\gamma = 1$, and this gives $a_2^+ = 5.3 \times 10^{-4}$ and $a_2^- = 2.8 \times 10^{-5}$.

In the healthy lung, only a small fraction of alveolar type II cells are actively proliferating at any one time (see ([20], fig. 1e), ([21], fig. 3k), and [54]). Assuming 1% of cells are proliferating, which means

$$1 - \frac{\bar{A}_1 + \bar{A}_2}{K_A} = 1\%.$$

Then we can estimate $K_A$ by

$$K_A = \frac{\bar{A}_1 + \bar{A}_2}{1 - 0.01} = 5.3 \times 10^4 \ 10^6 \text{cells}.$$

Moreover,

$$r_2 = \frac{\delta + \sigma_A}{1 - \dfrac{\bar{A}_1 + \bar{A}_2}{K_A}} = 0.055 \text{ h}^{-1}.$$

Note this gives an average cell cycle time of 18 h, which is somewhat shorter than that previously estimated (22 h), but biologically feasible, given the duration of synthesis phase (S) through mitosis phase (M) is approximately 9 h [54].

## A.2. Parameters describing the demographics and actions of alveolar macrophages and infiltrating immune cells

Our model includes both resting and active innate immune cells. We focus our attention on macrophages and neutrophils, where alveolar macrophages are imagined as the predominant phagocyte in the resting lung and neutrophils the predominant phagocyte during the early stages of infection. Since our model only distinguishes between innate immune cells in terms of their activation status, parameters describing resting immune cells are informed by studies of alveolar macrophages, while parameters describing active immune cells are informed by studies of neutrophils. However, in some cases, parameter values will be adjusted to reflect the expected ratio of these two cell types.

The lifespan of a human alveolar macrophage is estimated to be 81 days [107]. This gives $\sigma_M = 1/81 \text{ d}^{-1} = 0.0005 \text{ h}^{-1}$. Since these cells are thought to be long-lived [108], we take $\sigma_M \in [0.0001, 0.0005] \text{ h}^{-1}$. There is considerable debate as to the lifespan of neutrophils within the blood and tissue, with some sources placing this number of the order of hours and others of the order of days [53,109]. Moreover, evidence suggests that multiple factors can alter the lifespan of a neutrophil in the context of infection [53]. Despite this complexity, it seems reasonable that, in the interest of limiting tissue damage [110] and in light of the tremendous number of neutrophils generated each day [53], activated neutrophils should not be long-lived. Hence, we assume that on average neutrophils undergo apoptosis within 2 days of activation, and set $\sigma_M^* = 1/2 \text{ d}^{-1} = 0.02 \text{ h}^{-1}$ with $\sigma_M^* \in [0.008, 0.05] \text{ h}^{-1}$.

The recruitment rate of alveolar macrophages into the healthy lung can be estimated as

$$r_M = \sigma_M \bar{M} = 3 \ 10^6 \text{ cells h}^{-1},$$

with $r_M \in [0.6, 3] \ 10^6 \text{ cells h}^{-1}$.

Insight into the rate of infiltration by innate immune cells in response to inflammation is provided by [59], where the concentration of macrophages and neutrophils in bronchoalveolar lavage (BAL) is measured post LPS challenge. Here, the initial concentration of macrophages was $5 \times 10^4$ macrophages ml$^{-1}$, while neutrophils were effectively absent. While neutrophil levels rapidly increased to a maximal concentration of approximately $25 \times 10^4$ neutrophils ml$^{-1}$ by day three, the macrophage response was significantly delayed. As we are interested in modelling the earliest stages of infection, our model of immune cell recruitment into the alveolar region is based on the increase in neutrophils observed in this study. In order to relate the concentration of cells in the BAL to the number of cells in the alveolar region, we roughly estimate that a concentration of $10^4$ cells per ml in the BAL from this study corresponds to a total of 1000 million cells in the lung. This conversion is based on the initial concentration of macrophages in the BAL [59] and the reported number of alveolar macrophages in the lung [58]. Hence, 25 000 million immune cells are expected to be recruited to the lung over a 3-day period. Meaning that $r_M^* \approx 350 \ 10^6 \text{ cells h}^{-1}$.

Infiltrating immune cells, such as macrophages and neutrophils, remove infectious cells and viral particles via efferocytosis and endocytosis/phagocytosis, respectively [111,112]. We were unable to

find measurements of the *per capita* rate at which macrophages ingest infectious cells. A reasonable approximation is offered by the rate at which macrophages engulf apoptotic neutrophils, which was quantified as a mass action process in [69], with a second-order rate constant of 0.54 ml/$10^6$ cells h for resting macrophages and 2.4 ml/$10^6$ cells h for activated macrophages.

Clearance of SARS-CoV-2 by professional phagocytes is not well characterized. However, *in vitro* studies suggest that human macrophages can help to clear SARS-CoV-1 [113,114]. In particular, these cells were observed to phagocytose SARS-CoV-1 virons [113] and were found to be largely refractory to SARS-CoV-1 infection, producing little to no viral progeny post infection [113,114], while mounting a robust inflammtory reponse [114,115]. An estimate of the second-order rate constant at which innate immune cells clear SARS-CoV-1 virons is given by the *in vitro* rate at which neutrophils clear *Staphylococcus epidermidis*, 1.2 ml/$10^6$ cells h [116]. This value seems reasonable, as it is on the same order of magnitude as the rate of macrophage efferocytosis.

When extrapolating *in vivo* clearance rates from *in vitro* data, we must consider how the surface to volume ratio varies between the *in vitro* and *in vivo* conditions. This ratio can impact the rate of phagocytosis/efferocytosis since phagocytes are expected to crawl along the epithelial surface. For example, in [69], adherent macrophages were cultured in 0.5 ml wells with apoptotic cells. Assuming a well surface area of 0.7 $cm^2$ [117], the surface to volume ratio is approximately 1.4 $cm^2$ per ml, which yields a two-dimensional, second-order rate constant for efferocytosis of 0.8 $cm^2/10^6$ cells h, for resting macrophages and 3.4 $cm^2/10^6$ cells h for active macrophages. This two-dimensional rate constant is consistent with the reported diffusion coefficient for the macrophage: 11 $\mu m^2$ $min^{-1}$ = $0.66 \times 10^{-9}$ $m^2 h^{-1}$ [118]. In particular, given this diffusion rate, $10^6$ macrophages would be expected to cover approximately $0.66 \times 10^{-3}$ $m^2 h^{-1}$ = 6.6 $cm^2 h^{-1}$, giving a two-dimensional second order rate constant for clearance of 6.6 $cm^2/10^6$ cells h. Now the alveolar surface is covered by a very thin layer of fluid, in which we expect viral particles to be suspended. The volume of this fluid (both lungs) is estimated at about 40 ml [32]. Meanwhile the surface area of the alveolar region is about 100 $m^2$ [32,58]. Thus, the ratio of surface area to volume in the alveolar region is of the order of $10^4$ $cm^2$ $ml^{-1}$. A two-dimensional rate constant of 1–10 $cm^2/10^6$ cells h then translates to a three-dimensional constant of about $10^{-4}$–$10^{-3}$ ml/$10^6$ cells h in the alveolar region of the lung. In light of these considerations, we take the second order rate constants for clearance of dead cells and virus by resting and active innate immune cells as $k_{M0} = 10^{-4}$ ml/$10^6$ cells h and $k_M = 3 \times 10^{-4}$ ml/$10^6$ cells h, respectively.

## A.3. Parameters describing viral dynamics

There exist a number of papers providing insights into the kinetics with which SARS-CoV-2 induces epithelial cells to produce new virons [60–62,68,90]. A potential limitation of this data is that it is usually derived from the infection of Vero cells, as opposed to alveolar type II cells. Evidence suggests that *in vitro* SARS-CoV-2 induced cells to release new virons into the extracellular space within 6 h of infection [60,61] (see figure 2*a,b* and figure 2*a*, respectively). In this regard, SARS-CoV-2 appears to be similar to SARS-CoV-1 and vesicular stomatitis virus (VSV) [119,120]. Although viral replication is a multi-step process, consisting of viral entry, viral replication, and viral release [62], our simple model describes the replication cycle as a single step, so that cells move directly between the susceptible and infectious class. We select parameters such that the maximal possible rate of transition between these two states is $\beta = 1/6 \, h^{-1}$, which means that, on average and in the presence of saturating concentrations of virus, 6 h will elapse between exposure to the virus and the release of new virus.

Once virus is introduced into the alveolar region, a fraction of cells will become infectious. This infection process begins with the interaction between the virus spike protein and the host cell's ACE2 receptor. As virus-cell contact and viral entry occur on faster time scales (of the order of seconds or minutes [62,121]) than viral replication and egression, we assume the fraction of cells that are transitioning to an infectious state is at instantaneous equilibrium with the concentration of virus and susceptible cells in such a way that the transition rate saturates as the fraction of receptors in complex with viral spike protein approaches one. Specifically, we assume that saturates as

$$f\left(A_2^+, V, K_V, \frac{q_V}{C_1}\right) = \frac{V}{V + q_V(A_2^+/2C_1) + K_V}, \tag{A 1}$$

where $q_V = N_R/N_S$, $N_R$ is the number of ACE2 receptors per cell, $N_S$ is the number of viral spike proteins and $C_1$ is the volume of the alveolar fluid. In addition, $K_V = K_{AI}/N_S$, where $K_{AI}$ is the dissociation constant for the SARS-CoV-2 spike protein-ACE2 receptor.

Estimates of the dissociation constant for the SARS-CoV-2 spike protein receptor binding domain (RBD)-ACE2 receptor complex vary widely with experimental method. Estimates range from 1 to 403 nM, [100,122–124] or $0.602 - 241.88 \times 10^{12}$ particles ml$^{-1}$. It is suggested that the affinity of the full spike protein trimer for the ACE2 dimer could be even greater [123]. Hence, we select a low-value $K_{AI} = 1 \times 10^5 \, 10^6$ particles ml$^{-1}$ in order to parametrize the model. After the other parameters are estimated, we will vary this value between $[0.2 \times 10^5, 2 \times 10^5] \, 10^6$ particles ml$^{-1}$ to reflect variability between SARS-CoV-2 strains. We assume approximately 100 spike proteins per viron/ACE2 receptors per alveolar type II cell, i.e. $N_R = N_S = 100$. This gives $K_V = 1 \times 10^3 \, 10^6$ pfu h$^{-1}$ and $q_V = 1$.

We are unable to find single-cell analysis of SARS-CoV-2 production rates; however, estimates from other infections are available. It is important to note that replication rates can vary across a population and throughout the infectious period by several orders of magnitude. For example, single cell replication rates for vesticular stomatitis virus can vary between 10 and 3000 pfu h$^{-1}$ [119]. Similarly, infectious cells were found to produce between 1 and 970 pfu 12 h post influenza A infection (H1N1 A) [65], which corresponds to average *per capita* production rates of 0.083–80.83 pfu h$^{-1}$ cell. As H1N1 and SARS-CoV-2 are both respiratory viruses, H1N1 plaque production rates may provide a reasonable estimate of SARS-CoV-2 production rates. For example, H1N1 and SARS-CoV-2 were found to replicate with similar dynamics in lung tissue [64]. Hence we initiate $\rho_V = 10$ pfu h$^{-1}$ cell, and use viral load data to fit $\rho_V$ in [1, 100].

SARS-CoV-2 has been reported to have a half-life about 1–7 h on surfaces at 21–23°C and 40–65% relative humidity [62,63]. On the other hand, from supplementary fig. 2 of [64], we estimate the rate of thermal inactivation during *in vitro* culture at 37°C to be 0.1–0.2 h$^{-1}$. Hence, we set $\sigma_V = 1/3$ h$^{-1}$ with $\sigma_V \in [0.1, 1/3]$ h$^{-1}$.

## A.4. Parameters describing the dynamics and actions of cytokines and chemokines

Generally speaking, cytokines and chemokines have short half-lives [125]. For example, in plasma the half-life of CXCL10, which may be involved in the pathogenesis of COVID-19 [126,127], may be less than 5 min [85]. It is important to note, however, that such estimates may vary with experimental methods, including the route of administration, method of quantifying decay and context (e.g. *in vitro* versus *in vivo*). Furthermore, factors expressed on the cell surface and extracellular matrix specifically extend the half-life of chemokines within inflamed tissues [84–86]. In the light of these considerations, we take the rate of decay of chemokines as $\sigma_X = 1$ h$^{-1}$, with $\sigma_X \in [0.5, 2]$, so that these proteins last on average 0.5 to 2 h in the alveolar region.

The half-lives of type I interferons (e.g. IFN-$\alpha$ and IFN-$\beta$) may be somewhat longer than those of other cytokines. For example, [77] places the half-life of IFN-$\alpha$ at 4–16 h and the half-life of IFN-$\beta$ at 1–2 h [77]. Alternatively, [128] reports the half-life of IFN-$\beta$ as 5 h. We set the decay rate of interferons to be $\sigma_F = \ln2/2 = 0.35$ h$^{-1}$ (i.e. the half-life is 2 h) with $\sigma_F \in [0.04, 0.69]$ h$^{-1}$.

*In vitro* experiments suggest that the anti-viral effects of interferon stimulation last approximately 7–10 days in the context of SARS-CoV-2 infection [67]. Therefore, we assume cells leave the interferon-stimulated class at a rate of $\mu = 1/8.5$ d$^{-1} = 0.005$ h$^{-1}$ with $\mu \in [0.004, 0.006]$ h$^{-1}$.

We use data from [76] for estimating the average rate of IFN-$\beta$ production in a human alveolar epithelial cell line (A549). Specifically, IFN-$\beta$ levels were observed to peak approximately 16 h post infection at 21 000 pg ml$^{-1}$. Since the cell density was $0.25 \times 10^6$ cells ml$^{-1}$ and IFN $-\beta$ is $2.7 \times 10^8$ U mg$^{-1}$ [75], we estimate an average IFN $-\beta$ production rate post-infection to be 525 pg/$10^6$ cells h$^{-1} \approx 142$ U/$10^6$ cells h$^{-1}$. Select immune cells may produce type I IFNs at a greater rate. For example, dendritic cell precursors were found to produce type I IFNs at an average rate of approximately 1 U/cell/day [74]. We note, however, that because these cells represent a small fraction of total peripheral blood mononuclear cells (PBMC), the average *per capita* production rate of this cell population was only about $1.2 \times 10^{-3}$ U/cell/day or about $0.05 \times 10^{-3}$ U/cell h$^{-1}$. In our model equations, we will measure type I interferons in pmol. We set the rate of production by the mixed innate immune cell population as $\rho_{F_1} = 0.05 \, 10^3$ U/$10^6$ cell h. Meanwhile, the rate of production by epithelial cells is $\rho_{F_2} = 0.142 \, 10^3$ U/$10^6$ cell h. Assuming the type I interferon is IFN $-\beta$, $2.7 \times 10^8$ U mg$^{-1}$ (with a range of $2.70–4.00 \times 10^8$ U mg$^{-1}$ IFN-$\beta$) [75,129], and a molecular mass of 18 kDa [75], we have $\rho_{F_1} = 0.01$ pmol/$10^6$ cell h, and the rate of production by epithelial cells would be $\rho_{F_2} = 0.03$ pmol/$10^6$ cell h.

Per cell cytokine production rates vary with cytokine and cell type over a range of approximately 0.1–100 molecules/cell s$^{-1}$ [70]. Hence we fix $\rho_X = 0.006$ with $\rho_X \in [0.0006 - 0.6]$ pmol/$10^6$ cell h$^{-1}$.

Several studies have characterized the kinetics of the antiviral response to interferon stimulation. It seems this response is rapid and robust, with cells achieving an antiviral state within approximately

two hours of signalling and in response to pico-molar concentrations of type I interferons [130,131], alternatively, to concentrations as low as 1 U ml$^{-1}$ of antiviral cellular secretions [76]. Moreover, [90] supports the idea that SARS-CoV-2 is an 'interferon-sensitive' virus, similar to VSV. Our estimates of the parameters that determine the rate at which cells transition to an antiviral state are informed by [76], where the timing of the antiviral response was measured in the context of VSV infection. We assume that at the maximal rate of transition (i.e. at the rate achieved post stimulation with saturating concentrations of type I IFNs), approximately 70% of cells will achieve an antiviral state within 2 h of stimulation. This gives $\alpha = 0.6 \, \text{h}^{-1}$, and is roughly consistent with fig. 5A of [76].

The fraction of cells that are transitioning to an antiviral state is modelled as

$$f(A, F; K_F, q_F) = \frac{F}{F + q_F \dfrac{A}{2} + K_F},$$

where $A$ is the total concentration (in pM) of alveolar cells that are not yet stimulated by interferons. That is, $A = (A_2^+ + A_2^- + A_1 + I)/C_1(10^{-2}/6.02)$, where $10^{-2}/6.02$ is the conversion factor, converting units of $10^6$ cells ml$^{-1}$ to pM.

In order to estimate the parameters $K_F$ and $q_F$, we first note that *in vitro*, where the cellular concentration is very low, the IFN concentration at which the antiviral response is half-maximal (antiviral EC$_{50}$) is probably determined by the dissociation constant between the IFN protein and receptor. *In vitro*, the antiviral EC$_{50}$ of type I interferons is extremely small. According to one source, it is of the order of just 1 U ml$^{-1}$ [90,132]. Another source has the antiviral EC$_{50}$ of IFN-$\alpha$ as approximately one picomolar and that of IFN-$\beta$ as a fraction of a picomolar [130]. However, the dissociation constants between IFN-$\alpha$ or IFN-$\beta$ and the IFN receptor ($K_{FR}$) are considerably larger; $K_{FR} = 0.4$–5 nM for IFN-$\alpha$ and $K_{FR} = 0.1$ nM for IFN-$\beta$ [130], i.e. EC$_{50} \ll K_{FR}$. We interpret this to mean that the *in vitro* antiviral response is hypersensitive. Hence, it may be that the antiviral response rate is also hypersensitive, i.e. that $K_F \ll K_{FR}$. On the other hand, the antiviral EC$_{50}$ is typically measured as an endpoint assay performed several hours post stimulation, for example, by measuring the fraction of cells that are protected from the cytopathic effects of infection. As such, the empirical antiviral EC$_{50}$ may differ from $K_F$, which is the concentration at which the *rate* of transition to the antiviral state is half maximal. In agreement with this idea, fig. 5 of [76] indicates that *in vitro* the rate at which cells transition to an antiviral state is not yet saturated with respect to IFN $-\alpha$ at a concentration of 64 U ml$^{-1}$, so that an antiviral EC$_{50}$ of the order of 100 U ml$^{-1}$ would seem reasonable for the response rate. While this value is greater than the anitviral EC$_{50}$ determined by cytopathic effect, it is similar to the dissociation constants between IFN and IFN receptor reported above. In light of these considerations, we let $K_F = 100$ pM. Finally, assuming that cells typically express only a small number of interferon receptors, and that limited receptor engagement is required to transduce the antiviral response, we set $q_F = 40$ [130].

Research on SARS-CoV-1 suggests that post SARS-CoV-2 infection alveolar type II cells and immune cells produce chemokines (e.g. CCL2 and CXCL10), thereby attracting additional immune cells to the site of infection [57,114,115]. We assume the concentration of chemokine required to induce a half-maximal chemotactic response (EC$_{50}$) is of the same order of magnitude as the dissociation constant for chemokine and chemokine receptor/chemokine-receptor-expressing cells. Indeed, this appears to be the case for CCL2 and monocytes *in vitro*, where the dissociation constant is 0.77 nM and the EC$_{50}$ for chemotaxis is 0.5 nM [71]. Meanwhile, CXCL10 was found to bind to its receptor, CXCR3, with a dissociation constant of 0.2–0.3 nM [73], and CCR1, another chemokine receptor, was found to bind its ligands with dissociation constants ranging from 70 pM to 2 nM [72]. In light of this data, we set $K_X = 500$ pM. Our approach to modelling the chemotactic response is different from that used to model the transition to the antiviral state. In particular, as the concentration of leukocytes in the blood is several orders of magnitude lower than the concentration of epithelial cells in the alveolar region (being of on the order of just $10^6$–$10^7$ cells ml$^{-1}$ [116]), and hence, several orders of magnitude lower than the dissociation constants between chemotaxis receptors and ligands, we assume that the dissociation constant is the primary determinant of the saturation of the rate of chemotaxis, and model the rate of chemotaxis as $\rho_X \, g(X; K_X)$.

## A.5. Parameters describing the production and actions of toxins

Infiltrating immune cells are a potential source of inflammation in the context of SARS-CoV-2 infection. In particular, these cells promote inflammation by releasing cytokines and toxic antimicrobial substances (e.g. histones and oxidants) into the extracellular space [33,34,81,133]. Neutrophils, in particular, are

known to release toxins through both degranulation and NETosis [34], the latter of which is the expulsion of neutrophil extracellular traps (NETs) including tissue-damaging histones and proteases [81,109]. NETs may trap, neutralize, or destroy pathogens [81,109]; however, NETs may also damage host cells [81,133–135]. Indeed, NETs have been shown to induce apoptosis of epithelial cells [136] and promote thrombus formation, which may result in tissue damage [135]. Multiple modes of NETosis have been described. One mode, vital NETosis, rapidly initiates gradual DNA leakage, while another, suicidal NETosis, results in the delayed but singular expulsion of a cell's total DNA content and concomitant cell death [109]. Several studies have recently characterized NET production in the context of SARS-CoV-2 infection [134,137,138]. In particular, [134] supports a model in which SARS-CoV-2 directly induces neutrophils to undergo protein arginine deiminase 4 (PAD4)-dependent NETosis [139]. Moreover, evidence suggests that while PAD-4 is essential for suicidal NETosis [109], it is dispensable for vital NETosis [140]. For this reason, we assume SARS-CoV-2 induces suicidal NETosis, resulting in cell death.

It appears that significant interaction with pathogens (e.g. infection, phagocytosis, or binding to cell surface receptors) is required for NETosis. In support of this assumption, neutrophils were observed to phagocytose *Candida albicans* prior to releasing NETs, and expression of SARS-CoV-2 antigens correlates strongly with NET production *in vitro* [134]. For concreteness, we assume that immune cells will only produce toxic NETs after phagocytosis of virus or infectious cells, that is, only fully active immune cells produce NETs. Letting half of NETotic cells complete the process in three hours, and supposing a fraction of active immune cells (50%) undergo NETosis, we set $\rho_T = \ln(2)/6 \ \mathrm{h}^{-1} \approx 0.12 \ \mathrm{h}^{-1}$.

The extent of NET-induced cell death is determined by the proportion of the alveolar surface that is in contact with NETs. We estimate this proportion using information from [136] and [141]. In response to SARS-CoV-2, neutrophils produce long strands of NETs with an average length of 50 μm [136]. Images of NETs from [136] and [141], suggest individual neutrophils produce many NET strands. Given the alveolar surface of a single lung is about 50 m$^2$ [58], and treating each NET as a circular disc with a radius of 50 μm, it would take roughly $6 \times 10^9$ NETs to cover the entire alveolar surface. We assume a small fraction (5%) of a cell's surface needs be covered to induce maximal death, so that the half-saturation constant for the rate of NET-induced cell death is $3 \times 10^8$ NETs. Similar to cells, we measure NETs in units of $10^6$ and take $K_T = 3 \times 10^2 \ 10^6$ NETs.

Meanwhile, evidence suggests that NETs induce multiple modes of cell death in epithelial cells, including pyroptosis, apoptosis and necrosis [82]. In general, the time to initiate cell death is context-dependent, occurring hours or days post-exposure [142]. However, evidence suggests that in response to NETs and toxic NET components, epithelial cells undergo death in a matter of hours [82,83,134]. We take $r = 0.1$, so that the average time to die post NET exposure is 10 h.

Note that we do not include NET-induced removal of virus in this model, since the impact of NETs on SARS-CoV-2 has not been established. Although NETS are able to trap and/or neutralize some viruses, others are resistant to the effects of NETS [109,143].

NETs may be degraded by DNase [87] or endocytosed by macrophages [88,144]. As DNase is weakly expressed in the lung tissue [145], and evidence suggests that DNase may be insufficient to completely clear NETs under physiological conditions [88,138,146], we focus on clearance by immune cells. We are unable to find estimates of the rate of NET clearance by immune cells, so we take these parameters as identical to those for the clearance of other dead cells.

## Appendix B. Figures

(see figures 8, 9, 10, 11, 12).

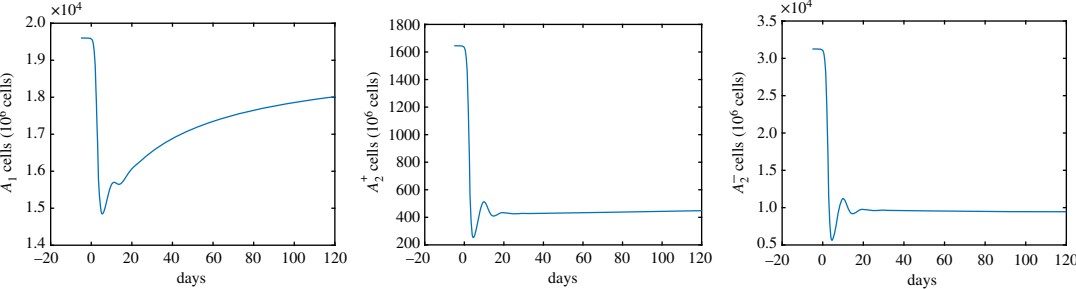

**Figure 8.** Simulated epithelial cell populations post infection over 60 days with parameters and initial values from tables 2 and 3.

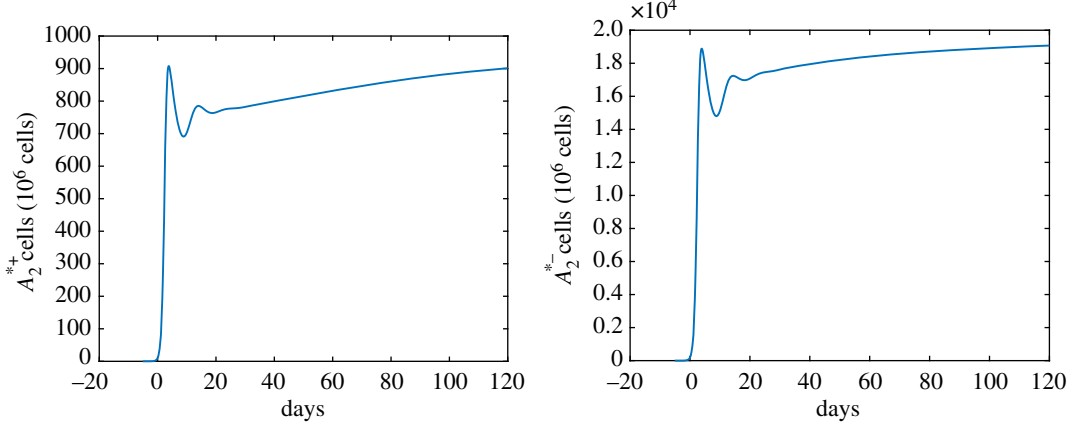

**Figure 9.** Simulated interferon-stimulated epithelial cell populations post infection over 60 days with parameters and initial values from tables 2 and 3.

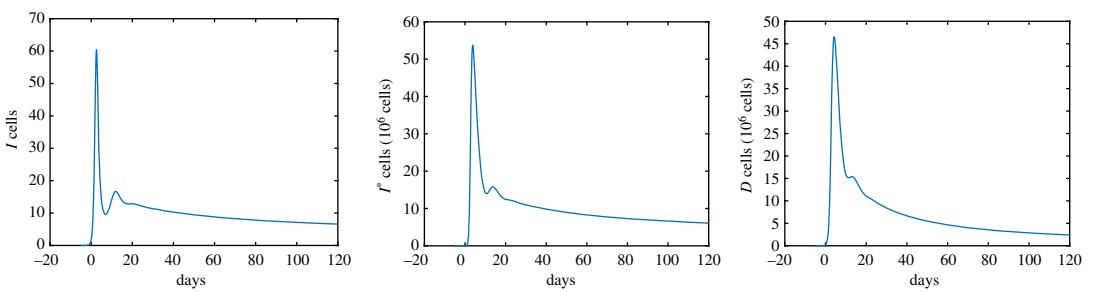

**Figure 10.** Simulated infectious epithelial cell populations post infection over 60 days with parameters and initial values from tables 2 and 3.

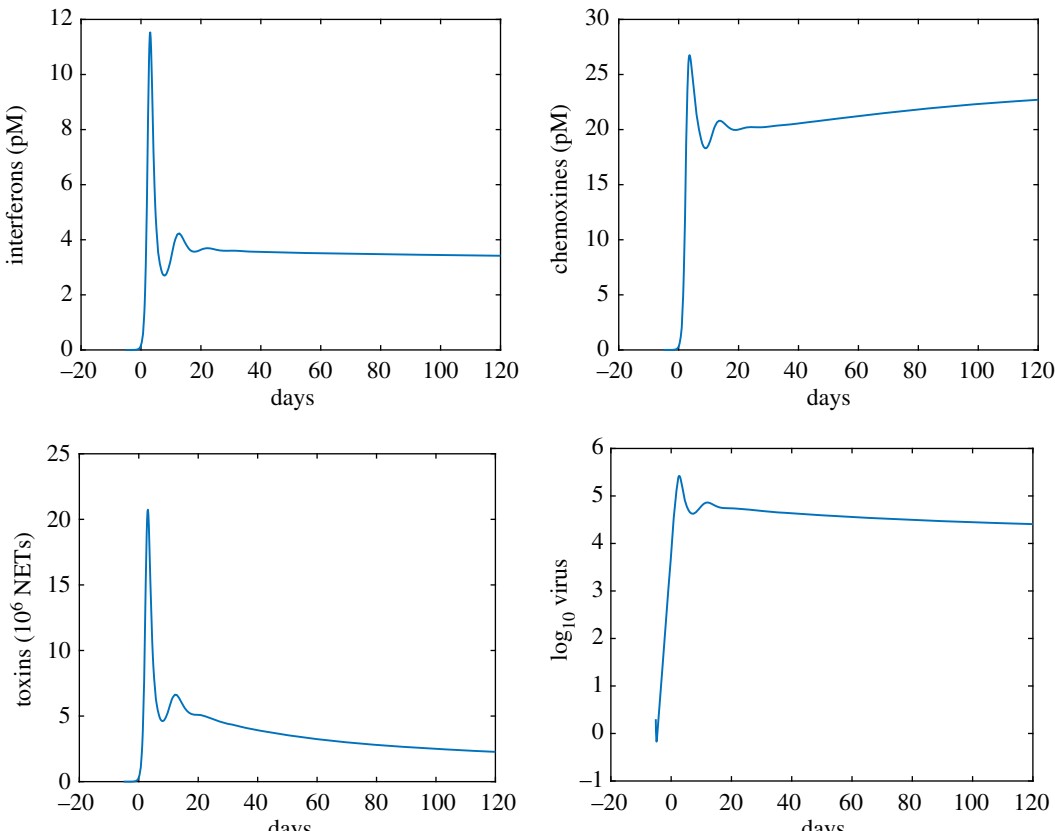

**Figure 11.** Simulated small molecules post infection over 60 days with parameters and initial values from tables 2 and 3.

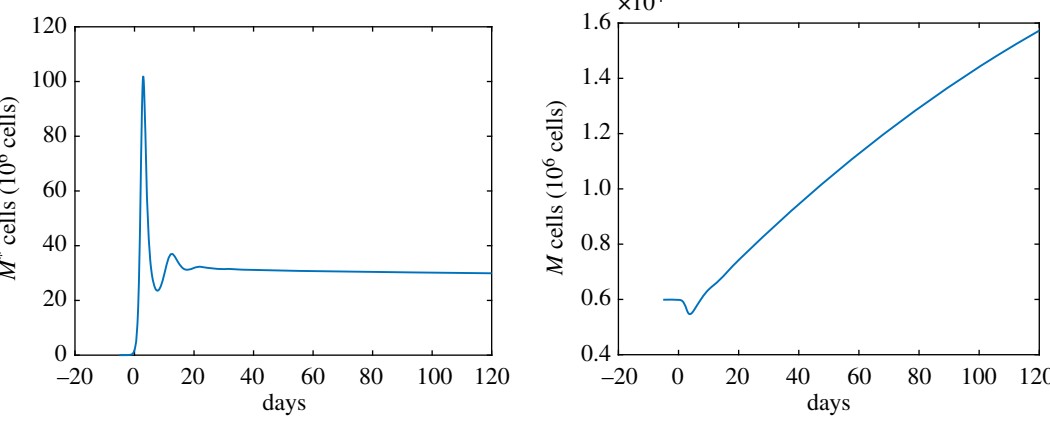

**Figure 12.** Simulated innate immune cells over 60 days with parameters and initial values from tables 2 and 3.

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
