## [Peer Review File · Royal Society Open Science]

Review History

RSOS-210090.R0 (Original submission)

Review form: Reviewer 1

Is the manuscript scientifically sound in its present form?

Yes

Are the interpretations and conclusions justified by the results?

No

Is the language acceptable?

Yes

Do you have any ethical concerns with this paper?

No

Have you any concerns about statistical analyses in this paper?

No

Recommendation?

Major revision is needed (please make suggestions in comments)

Comments to the Author(s)

In this paper, the authors developed a system of differential equations to study the innate immune responses to SARS-CoV-2 infection. The model is relatively complex, including type I and II alveolar epithelial cells, interferon, chemokines, toxins, and innate immune cells. The authors derived the basic reproduction number and studied the stability of the disease-free equilibrium. By fitting the model to some viral load data, they obtained some parameter values, based on which they evaluated the efficiency of the innate immune response and explored the impact of potential interferon or antiviral treatment on the course of infection.

The authors provided a nice review of the within-host models for SARS-CoV-2 infection. Most of the models in the literature do not include the immune responses explicitly, possibly due to the lack of immune data. The authors introduced many components involved in the infection and developed the differential equation model. The model formulation seems reasonable.

As the model is relatively complex, my concern is whether such a model can be well validated by available data. Most of the parameters are chosen from the literature. A few of them are obtained by fitting the model to very limited viral load data (Figure 1). My concern is whether the predictions (e.g. the control of infection by the innate immune response and the impact of potential treatment) based on these parameter values are reliable. The authors may at least want to add some sensitivity tests on some key parameters and see how they might change the model prediction.

Minor issue: in the abstract, “a single viral particle produces on average much less than one infected cell”: within-host R_0 is the number of virions induced by one virion or the number of newly infected cells induced by one infected cell in a wholly susceptible environment.

Page 20, “we increase the viral loads in [87] by two orders of magnitude prior to fitting” any justification why it is two orders higher?

Review form: Reviewer 2 (Thomas Hillen)

Is the manuscript scientifically sound in its present form?

Yes

Are the interpretations and conclusions justified by the results?

Yes

Is the language acceptable?

Yes

Do you have any ethical concerns with this paper?

No

Have you any concerns about statistical analyses in this paper?

No

Recommendation?

Accept with minor revision (please list in comments)

Comments to the Author(s)

Thank you for letting me review this excellent paper by Leander et al. The authors carefully derive a mathematical model for viral infection of lung-alveolar cells. The alveolar epithelium is the main target of coronavirus infections, and a good understanding of its viral response is essential. Great care is taken in this paper to derive the mathematical model for type one and type two alveolar cells, their ACE2-receptor expression, interferon, and cytokine dynamics and the impact of the innate immune response. I wish all modelling papers would present such a level of detail and clarity as seen here. Also the parameter estimation, as presented in the appendix, is done wonderfully and quite convincing, basing model parameters on the available literature.

Considering the high quality of the modelling process, I find the presented analysis a bit thin. The authors invested quite some work into the modelling, hence I would have loved to see a more substantial model analysis. I am aware that there are millions of possible ways the study could be extended, and I am not requesting a full analysis of all cases. But still, I hope I can stimulate the authors to include one or two more cases that carry some weight. For example:

- (1) The model parameters allow ranges from low to high. Can this be used to somehow classify patients into low-risk, medium-risk, high-risk populations?
- (2) In the virus literature (see below), it is often discussed that the virus response can be bi-phasic or tri-phasic. I think your model should get the bi-phasic behavior very well.
- (3) Have you done an individual fit to the patient data from To2020?

As I said, I am not requesting you to do all of these. But it would be nice to add at least one substantial piece of analysis.

Some minor comments:

- Section 2: Please include a flow diagram for the mathematical model. This would help with the intuitive understanding of the model.
- Page 6 eq (2.1): Please say that $A_2 = A_{2plus} + A_{2minus}$
- Page 14: you end up with a 14-equation model. Please mention how many parameters are involved. (maybe around 60?)
- Page 22: you fit a 60-parameter model to 11 data points. Some explanation is needed.
- Figure 4: Close to the initial data you have a fast decrease of viral load before it starts to grow. This is not very realistic, and also, it cannot be supported by the data. Can you change your initial conditions such that you see immediate growth? Or is there a good reason for this decay? Then explain it.
- Figure 5,6 etc. The axis labels are way too small.

A paper where viral phases are explained:

Validated models of immune response to virus infection. Amber M. Smith, *Current Opinion in Systems Biology* 2018, 12:46–52

A paper where the same data from To are analysed, and the three viral phases are discussed:

Personalized Virus Load Curves of SARS-CoV-2 Infection, Thomas Hillen, Carlos Contreras, Jay M. Newby, medRxiv, doi: <https://doi.org/10.1101/2021.01.21.21250268>.

Decision letter (RSOS-210090.R0)

Dear Dr Leander

On behalf of the Editors, we are pleased to inform you that your Manuscript RSOS-210090 "A model of the innate immune response to SARS-CoV-2 in the alveolar epithelium" has been accepted for publication in Royal Society Open Science subject to minor revision in accordance with the referees' reports. Please find the referees' comments along with any feedback from the Editors below my signature.

Please submit your revised manuscript and required files (see below) no later than 7 days from today's (ie 15-Jun-2021) date. Note: the ScholarOne system will 'lock' if submission of the revision is attempted 7 or more days after the deadline. If you do not think you will be able to meet this deadline please contact the editorial office immediately.

on behalf of Dr Jianhong Wu (Associate Editor) and Glenn Webb (Subject Editor)
openscience@royalsociety.org

Reviewer comments to Author:

Reviewer: 1

Comments to the Author(s)

In this paper, the authors developed a system of differential equations to study the innate immune responses to SARS-CoV-2 infection. The model is relatively complex, including type I and II alveolar epithelial cells, interferon, chemokines, toxins, and innate immune cells. The authors derived the basic reproduction number and studied the stability of the disease-free equilibrium. By fitting the model to some viral load data, they obtained some parameter values, based on which they evaluated the efficiency of the innate immune response and explored the impact of potential interferon or antiviral treatment on the course of infection.

The authors provided a nice review of the within-host models for SARS-CoV-2 infection. Most of the models in the literature do not include the immune responses explicitly, possibly due to the lack of immune data. The authors introduced many components involved in the infection and developed the differential equation model. The model formulation seems reasonable.

As the model is relatively complex, my concern is whether such a model can be well validated by available data. Most of the parameters are chosen from the literature. A few of them are obtained by fitting the model to very limited viral load data (Figure 1). My concern is whether the predictions (e.g. the control of infection by the innate immune response and the impact of potential treatment) based on these parameter values are reliable. The authors may at least want to add some sensitivity tests on some key parameters and see how they might change the model prediction.

Minor issue: in the abstract, “a single viral particle produces on average much less than one infected cell”: within-host R_0 is the number of virions induced by one virion or the number of newly infected cells induced by one infected cell in a wholly susceptible environment.

Page 20, “we increase the viral loads in [87] by two orders of magnitude prior to fitting” any justification why it is two orders higher?

Reviewer: 2

Comments to the Author(s)

Thank you for letting me review this excellent paper by Leander et al. The authors carefully derive a mathematical model for viral infection of lung-alveolar cells. The alveolar epithelium is the main target of coronavirus infections, and a good understanding of its viral response is essential. Great care is taken in this paper to derive the mathematical model for type one and type two alveolar cells, their ACE2-receptor expression, interferon, and cytokine dynamics and the impact of the innate immune response. I wish all modelling papers would present such a level of detail and clarity as seen here. Also the parameter estimation, as presented in the appendix, is done wonderfully and quite convincing, basing model parameters on the available literature.

Considering the high quality of the modelling process, I find the presented analysis a bit thin. The authors invested quite some work into the modelling, hence I would have loved to see a more substantial model analysis. I am aware that there are millions of possible ways the study could be extended, and I am not requesting a full analysis of all cases. But still, I hope I can stimulate the authors to include one or two more cases that carry some weight. For example:

- (1) The model parameters allow ranges from low to high. Can this be used to somehow classify patients into low-risk, medium-risk, high-risk populations?
- (2) In the virus literature (see below), it is often discussed that the virus response can be bi-phasic or tri-phasic. I think your model should get the bi-phasic behavior very well.
- (3) Have you done an individual fit to the patient data from To2020?

As I said, I am not requesting you to do all of these. But it would be nice to add at least one substantial piece of analysis.

Some minor comments:

- Section 2: Please include a flow diagram for the mathematical model. This would help with the intuitive understanding of the model.
- Page 6 eq (2.1): Please say that $A_2 = A_{2plus} + A_{2minus}$
- Page 14: you end up with a 14-equation model. Please mention how many parameters are involved. (maybe around 60?)
- Page 22: you fit a 60-parameter model to 11 data points. Some explanation is needed.
- Figure 4: Close to the initial data you have a fast decrease of viral load before it starts to grow. This is not very realistic, and also, it cannot be supported by the data. Can you change your initial conditions such that you see immediate growth? Or is there a good reason for this decay? Then explain it.
- Figure 5,6 etc. The axis labels are way too small.

A paper where viral phases are explained:

Validated models of immune response to virus infection. Amber M. Smith, *Current Opinion in Systems Biology* 2018, 12:46–52

A paper where the same data from To are analysed, and the three viral phases are discussed:

Personalized Virus Load Curves of SARS-CoV-2 Infection, Thomas Hillen, Carlos Contreras, Jay M. Newby, medRxiv, doi: <https://doi.org/10.1101/2021.01.21.21250268>.

===PREPARING YOUR MANUSCRIPT===

===PREPARING YOUR REVISION IN SCHOLARONE===

Please ensure that you include a summary of your paper at Step 2 'Type, Title, & Abstract'. This should be no more than 100 words to explain to a non-scientific audience the key findings of your

research. This will be included in a weekly highlights email circulated by the Royal Society press office to national UK, international, and scientific news outlets to promote your work.

Author's Response to Decision Letter for (RSOS-210090.R0)

See Appendix A.

Decision letter (RSOS-210090.R1)

Dear Dr Leander,

I am pleased to inform you that your manuscript entitled "A model of the innate immune response to SARS-CoV-2 in the alveolar epithelium" is now accepted for publication in Royal Society Open Science.

COVID-19 rapid publication process:

We are taking steps to expedite the publication of research relevant to the pandemic. If you wish, you can opt to have your paper published as soon as it is ready, rather than waiting for it to be published the scheduled Wednesday.

This means your paper will not be included in the weekly media round-up which the Society sends to journalists ahead of publication. However, it will still appear in the COVID-19 Publishing Collection which journalists will be directed to each week (<https://royalsocietypublishing.org/topic/special-collections/novel-coronavirus-outbreak>).

If you wish to have your paper considered for immediate publication, or to discuss further, please notify openscience_proofs@royalsociety.org and press@royalsociety.org when you respond to this email.

on behalf of Dr Jianhong Wu (Associate Editor) and Glenn Webb (Subject Editor)
openscience@royalsociety.org

Appendix A

We thank the reviewers for their careful and thoughtful comments. We feel that the paper is significantly improved as a result of their efforts. Below, please find our point-by-point responses to the reviewer comments in blue.

Reviewer comments to Author:

Reviewer: 1

Comments to the Author(s)

In this paper, the authors developed a system of differential equations to study the innate immune responses to SARS-CoV-2 infection. The model is relatively complex, including type I and II alveolar epithelial cells, interferon, chemokines, toxins, and innate immune cells. The authors derived the basic reproduction number and studied the stability of the disease-free equilibrium. By fitting the model to some viral load data, they obtained some parameter values, based on which they evaluated the efficiency of the innate immune response and explored the impact of potential interferon or antiviral treatment on the course of infection.

The authors provided a nice review of the within-host models for SARS-CoV-2 infection. Most of the models in the literature do not include the immune responses explicitly, possibly due to the lack of immune data. The authors introduced many components involved in the infection and developed the differential equation model. The model formulation seems reasonable.

As the model is relatively complex, my concern is whether such a model can be well validated by available data. Most of the parameters are chosen from the literature. A few of them are obtained by fitting the model to very limited viral load data (Figure 1). My concern is whether the predictions (e.g. the control of infection by the innate immune response and the impact of potential treatment) based on these parameter values are reliable. The authors may at least want to add some sensitivity tests on some key parameters and see how they might change the model prediction.

It is true that the model's behavior is determined by its structure and parameters. The model's structure and parameters were thoroughly researched, and we provide a detailed description of how the model was constructed from the literature so that readers can understand our results in context. Model systems, be they numerical or empirical, are typically controlled and constrained so that results are interpretable. This research methodology provides insight into how parts behave in isolation. In order for the model to be useful, it should be comparable to other experiments and real-world data. Viral load and cell damage are some outputs of our model that are frequently measured in patient studies and animal models. Differences between our model outputs and data gathered from other model systems can help us understand how the innate immune response contributes to the overall immune response and evidence gaps or inaccuracies in our knowledge.

In this revision, we add three new numerical experiments to the paper in order to provide a more thorough exploration of model-predicted variability in disease severity. Specifically, we now consider how variability in ACE2 expression, epithelial cell interferon production, and binding affinity between SARS-CoV2 and ACE2 impact prognosis. These experiments provide a more complete view of the range of possible outcomes predicted by the model.

Minor issue: in the abstract, "a single viral particle produces on average much less than one infected cell": within-host R_0 is the number of virions induced by one virion or the number of newly infected cells induced by one infected cell in a wholly susceptible environment.

We agree with this definition of R_0 and note that it matches the definition in the paper. As explained on page 15 of the paper, in this model, R_0 is the geometric mean of the average number of cells infected by a single viral pfu and the average number of viral pfu produced by an infected cell in a susceptible population. The statement in the abstract only concerns the first of these factors.

Page 20, “we increase the viral loads in [87] by two orders of magnitude prior to fitting” any justification why it is two orders higher?

The exact factor by which the viral load should be converted is unknown. As explained on page 20 of the paper, we anticipate the viral load to be significantly greater in the alveolar fluid than it is in the saliva owing to differences in the rate of decay in these two compartments. For example, if we swallow on average once over two minutes throughout the day, the average rate of turnover of the saliva is about 30 1/hr. Meanwhile, we expect the rate of turnover in the alveolar epithelium to be dominated by thermal inactivation, which occurs at a rate of approximately .3 1/hr.

Reviewer: 2

Comments to the Author(s)

Thank you for letting me review this excellent paper by Leander et al. The authors carefully derive a mathematical model for viral infection of lung-alveolar cells. The alveolar epithelium is the main target of coronavirus infections, and a good understanding of its viral response is essential. Great care is taken in this paper to derive the mathematical model for type one and type two alveolar cells, their ACE2-receptor expression, interferon, and cytokine dynamics and the impact of the innate immune response. I wish all modelling papers would present such a level of detail and clarity as seen here. Also the parameter estimation, as presented in the appendix, is done wonderfully and quite convincing, basing model parameters on the available literature.

Considering the high quality of the modelling process, I find the presented analysis a bit thin. The authors invested quite some work into the modelling, hence I would have loved to see a more substantial model analysis. I am aware that there are millions of possible ways the study could be extended, and I am not requesting a full analysis of all cases. But still, I hope I can stimulate the authors to include one or two more cases that carry some weight. For example:

(1) The model parameters allow ranges from low to high. Can this be used to somehow classify patients into low-risk, medium-risk, high-risk populations?

Given that the model is confined to the initial innate immune response, and risk is influenced by many factors including chronic disease, we cannot confidently classify patients into risk groups based on the results of this model. However, as described in the response to reviewer one, we have added additional numerical experiments to quantify how individual variability influences prognosis.

(2) In the virus literature (see below), it is often discussed that the virus response can be bi-phasic or tri-phasic. I think your model should get the bi-phasic behavior very well.

Our simulations do match the biphasic chronic response described in the paper by Smith. We have added a note about this to page 20 of the paper.

(3) Have you done an individual fit to the patient data from To2020?

We have not fit the model to the individual data from this paper. Although we agree that this would be an interesting experiment, in this revision we have instead conducted additional numerical experiments to explore the impact of individual variability on prognosis.

As I said, I am not requesting you to do all of these. But it would be nice to add at least one substantial piece of analysis.

Some minor comments:

- Section 2: Please include a flow diagram for the mathematical model. This would help with the intuitive understanding of the model.

Because the model is so complex, it seemed challenging to create a readable flow diagram of the model.

- Page 6 eq (2.1): Please say that $A_2 = A_{2plus} + A_{2minus}$

Done

- Page 14: you end up with a 14-equation model. Please mention how many parameters are involved. (maybe around 60?)

We have 14 variables and 39 parameters. This is now mentioned on page 14.

- Page 22: you fit a 60-parameter model to 11 data points. Some explanation is needed.

We have improved the explanation of the method of model fitting on page 20 to explain that three parameter values are fit to the data. Other parameter values are fixed based on information from the literature.

- Figure 4: Close to the initial data you have a fast decrease of viral load before it starts to grow. This is not very realistic, and also, it cannot be supported by the data. Can you change your initial conditions such that you see immediate growth? Or is there a good reason for this decay? Then explain it.

This decay occurs because there is a delay of several hours before infected cells begin to produce new virus. During this time, the concentration of virus falls due to thermal inactivation. Note that data collection began several days after initial exposure, at the onset of symptoms. Hence, the initial dip in the viral load should not be present in the data.

- Figure 5,6 etc. The axis labels are way too small.

We enlarged the axis labels.

A paper where viral phases are explained:

Validated models of immune response to virus infection. Amber M. Smith, Current Opinion in Systems Biology 2018, 12:46–52

A paper where the same data from To are analysed, and the three viral phases are discussed:

Personalized Virus Load Curves of SARS-CoV-2 Infection, Thomas Hillen, Carlos Contreras, Jay M. Newby, medRxiv, doi: <https://doi.org/10.1101/2021.01.21.21250268>.

===PREPARING YOUR MANUSCRIPT===

- one version identifying all the changes that have been made (for instance, in coloured highlight, in bold text, or tracked changes);
- a 'clean' version of the new manuscript that incorporates the changes made, but does not highlight them. This version will be used for typesetting.

===PREPARING YOUR REVISION IN SCHOLARONE===

- If you are requesting a discretionary waiver for the article processing charge, the waiver form must be included at this step.
- If you are providing image files for potential cover images, please upload these at this step, and inform the editorial office you have done so. You must hold the copyright to any image provided.
- A copy of your point-by-point response to referees and Editors. This will expedite the preparation of your proof.

- Ensure that your data access statement meets the requirements at <https://royalsociety.org/journals/authors/author-guidelines/#data>. You should ensure that you cite the dataset in your reference list. If you have deposited data etc in the Dryad repository, please only include the 'For publication' link at this stage. You should remove the 'For review' link.
- If you are requesting an article processing charge waiver, you must select the relevant waiver option (if requesting a discretionary waiver, the form should have been uploaded at Step 3 'File upload' above).
- If you have uploaded ESM files, please ensure you follow the guidance at <https://royalsociety.org/journals/authors/author-guidelines/#supplementary-material> to include a suitable title and informative caption. An example of appropriate titling and captioning may be found at https://figshare.com/articles/Table_S2_from_Is_there_a_trade-off_between_peak_performance_and_performance_breadth_across_temperatures_for_aerobic_scope_in_teleost_fishes_/3843624.
